# Natural variation in the *Caenorhabditis elegans* egg-laying circuit modulates an intergenerational fitness trade-off

**Laure Mignerot[1†], Clotilde Gimond[1†], Lucie Bolelli[1], Charlotte Bouleau[1], Asma Sandjak[1], Thomas Boulin[2], Christian Braendle[1]\***

[1]Université Côte d'Azur, CNRS, Inserm, IBV, Nice, France; [2]Institut NeuroMyoGène, CNRS, Inserm, Université de Lyon, Lyon, France

**Abstract** Evolutionary transitions from egg laying (oviparity) to live birth (viviparity) are common across various taxa. Many species also exhibit genetic variation in egg-laying mode or display an intermediate mode with laid eggs containing embryos at various stages of development. Understanding the mechanistic basis and fitness consequences of such variation remains experimentally challenging. Here, we report highly variable intra-uterine egg retention across 316 *Caenorhabditis elegans* wild strains, some exhibiting strong retention, followed by internal hatching. We identify multiple evolutionary origins of such phenotypic extremes and pinpoint underlying candidate loci. Behavioral analysis and genetic manipulation indicates that this variation arises from genetic differences in the neuromodulatory architecture of the egg-laying circuitry. We provide experimental evidence that while strong egg retention can decrease maternal fitness due to in utero hatching, it may enhance offspring protection and confer a competitive advantage. Therefore, natural variation in *C. elegans* egg-laying behaviour can alter an apparent trade-off between different fitness components across generations. Our findings highlight underappreciated diversity in *C. elegans* egg-laying behavior and shed light on its fitness consequences. This behavioral variation offers a promising model to elucidate the molecular changes in a simple neural circuit underlying evolutionary shifts between alternative egg-laying modes in invertebrates.

## eLife assessment

This **important** work provides a thorough and detailed analysis of natural variation in *C. elegans* egg-laying behavior. The authors present **convincing** evidence to support their hypothesis that variations in egg-laying behavior are influenced by trade-offs between maternal and offspring fitness. This study establishes a framework for elucidating the molecular mechanisms underlying this paradigm of behavioral evolution.

## Introduction

Reproductive strategies reflect higher-order phenotypes that emerge through integration of developmental, morphological, physiological, and behavioural phenotypes. Understanding how these different phenotypes integrate and co-evolve to explain the diversification in reproductive strategies is a central goal of life history research (*Flatt and Heyland, 2011*). One such fundamental aspect of reproduction in internally fertilizing animals is the duration of intra-uterine embryonic development, which can be highly variable and is influenced by an interplay of genetic and environmental factors. While oviparity and viviparity reflect opposite extremes, many taxa display intermediate modes by laying eggs containing embryos at variably advanced stages of development (sometimes termed

**\*For correspondence:**
braendle@unice.fr

[†]These authors contributed equally to this work

**Competing interest:** The authors declare that no competing interests exist.

ovoviviparity). The degree of such maternal retention of fertilized eggs may vary significantly between closely related species or even between genotypes of the same species, for example in insects, snails, or reptiles, and is further influenced by diverse environmental factors, such as mating status or food availability (*Meier et al., 1999*; *Markow et al., 2009*; *Bleu et al., 2012*; *Kalinka, 2015*). Viviparity – and prolonged egg retention in general – are thought to promote maternal control over the offspring environment, providing an extended opportunity for offspring resource provisioning as well as improved offspring protection against environmental fluctuations, stressors, pathogens or predators (*Blackburn, 1999*; *Avise, 2013*; *Kalinka, 2015*; *Ostrovsky et al., 2016*). In addition, laying of advanced-stage embryos or larvae may generate an advantage during intra- and interspecific competition by reducing external egg-to-adult developmental time, for example, in insects (*Bakker, 1961*; *Kalinka, 2015*; *Mueller and Bitner, 2015*). Despite these apparent benefits, prolonged egg retention and viviparity are generally presumed to incur fitness costs, expressed as reduced maternal survival and fecundity, for example, due to higher metabolic costs as a result of prolonged gestation (*Kalinka, 2015*). How variation in ecological niche drives the evolution of different egg-laying modes given these apparent trade-offs has been a major focus of evolutionary ecological research (*Blackburn, 1999*; *Avise, 2013*; *Kalinka, 2015*; *Ostrovsky et al., 2016*).

The study of evolutionary transitions between ovi- and viviparity has concentrated on comparisons between species, so that evidence is mainly correlative and the genetic changes underlying such transitions can only rarely be determined (*Horváth and Kalinka, 2018*; *Recknagel et al., 2021*). Relatively few studies have examined quantitative intraspecific variation in egg retention although this approach may facilitate disentangling its relative costs and benefits. Intraspecific analysis may further allow for identification of genomic loci contributing to this variation and potentially provide insights into the molecular changes during incipient stages of evolutionary transitions towards obligate viviparity. One such study was performed for natural quantitative variation in egg retention of fertilized *Drosophila* females (*Horváth and Kalinka, 2018*): Genome-wide association (GWA) mapping yielded 15 potential candidate genes harbouring natural polymorphisms contributing to variation in egg retention. These variants have not yet been validated and it remains unclear which particular traits, such as fecundity or egg-laying behaviour, contribute to observed variation in egg retention (*Horváth and Kalinka, 2018*). In this study, prolonged *Drosophila* egg retention was correlated with a reduction in fecundity, in line with the prediction that evolutionary transitions towards viviparity impose fitness costs (*Clutton-Brock, 1991*; *Blackburn, 1999*; *Avise, 2013*; *Kalinka, 2015*; *Ostrovsky et al., 2016*). Analysis of intraspecific variation may thus be valuable to determine genetic causes and fitness consequences of divergent egg retention, providing an entry point into understanding evolutionary transitions between ovi- and viviparity.

In our study, to characterize intraspecific differences in egg retention, we focused on natural variation in egg laying of the male-hermaphrodite (androdioecious) *Caenorhabditis elegans*. In optimal conditions, this primarily self-fertilizing nematode completes its life cycle within 3–4 days, and hermaphrodites generate around 150–300 offspring over a period of 2–3 days. During the reproductive phase, animals of the reference strain (N2) accumulate up to ~15 fertilized eggs in the uterus, caused by a delay between ovulation and egg laying (*Schafer, 2006*). Embryonic development takes around 15 hours and occurs independently of whether eggs are laid or retained in the uterus. On average, embryos spend two to three hours in utero, reaching the early gastrula (~30-cell) stage when they are being laid (*Sulston et al., 1983*). At any given time, the number of eggs retained in utero will therefore depend on rates of both egg production and egg laying. Reduced egg-laying rates lead to the accumulation of advanced stage embryos in utero. Egg laying is a rhythmic behaviour that alternates between inactive (~20min) and active (~2min) phases (*Waggoner et al., 1998*), regulated by a structurally simple neural circuit (*White et al., 1986*). During active egg-laying phases, the neurotransmitter serotonin and neuropeptides signal via the hermaphrodite-specific neurons (HSN) to increase excitability of vulva muscle cells, causing the rapid sequential release of four to six eggs (*Waggoner et al., 1998*; *Shyn et al., 2003*; *Zhang et al., 2008*; *Collins et al., 2016*; *Brewer et al., 2019*). Additional neuromodulatory signals and mechanosensory feedback regulate temporal patterns of egg-laying activity, which are further modulated by physiological and environmental inputs (*Trent et al., 1983*; *Schafer, 2005*; *Ringstad and Horvitz, 2008*; *Koelle, 2018*; *Fernandez et al., 2020*; *Ravi et al., 2021*; *Aprison et al., 2022*; *Medrano and Collins, 2023*). Food signals are required for sustained egg-laying activity (*Horvitz et al., 1982*; *Trent, 1982*; *Waggoner et al., 1998*; *Daniels et al., 2000*;

*Dong et al., 2000*), whereas diverse stressors, such as starvation, hypoxia, thermal stress, osmotic stress, or pathogens, inhibit egg laying (*Trent, 1982*; *Aballay et al., 2000*; *Waggoner et al., 2000*; *Chen and Caswell-Chen, 2003*; *Schafer, 2005*; *Zhang et al., 2008*; *McMullen et al., 2012*; *Fenk and de Bono, 2015*). Prolonged stress exposure will lead to intra-uterine egg retention and internal (matricidal) hatching, so that larvae develop inside their mother, often leading to premature maternal death (*Maupas, 1900*; *Trent, 1982*; *Chen and Caswell-Chen, 2003*; *Chen and Caswell-Chen, 2004*). Environmentally induced egg retention and internal hatching in *C. elegans* can be considered a plastic switch from oviparity to viviparity, also termed facultative viviparity (*Chen and Caswell-Chen, 2004*). In *C. elegans*, while eggs can no longer be provisioned after fertilization, larvae in utero may feed on decaying maternal tissues allowing them to develop into advanced larval stages, even in nutrient-scarce environments (*Chen and Caswell-Chen, 2004*). Stress-induced internal hatching in *C. elegans* may thus reflect adaptive phenotypic plasticity allowing for offspring provisioning, in particular, by enabling larvae to develop into the diapausing, starvation-resistant dauer larval stage (*Chen and Caswell-Chen, 2004*), which also represents the key dispersal developmental stage, able to colonize novel favourable environments (*Félix and Braendle, 2010*).

The detailed genetic understanding of *C. elegans* egg-laying behaviour has been obtained through the study of a single wild-type genotype, the laboratory strain N2, and its mutant derivatives (*Schafer, 2006*). It therefore remains unclear to what extent the egg-laying circuit harbours intraspecific variation. Although wild strains are highly isogenic due to a predominant self-fertilization (selfing) (*Barrière and Félix, 2005*; *Lee et al., 2021*), *C. elegans* retains considerable genetic (and phenotypic) diversity across the globe, with strains differing genetically, often in localized, highly divergent genomic regions (*Andersen et al., 2012*; *Crombie et al., 2019*; *Lee et al., 2021*; *Gilbert et al., 2022*). While egg-laying responses to diverse stimuli seem to be largely invariant across many *C. elegans* populations (*Chen and Caswell-Chen, 2004*; *Chen et al., 2020*; *Vigne et al., 2021*), wild strains may exhibit strongly divergent egg-laying behaviour (*Vigne et al., 2021*). These rare strains display constitutively strong egg retention (up to ~50 eggs in utero) and internal hatching irrespective of the environment (partial viviparity). A single, major-effect variant explains this derived phenotype: a single amino acid substitution (V530L) in KCNL-1, a small-conductance calcium-activated potassium channel subunit. This gain-of-function mutation causes vulval muscle hyperpolarization to reduce egg-laying activity, leading to constitutively strong egg retention and internal hatching (*Vigne et al., 2021*). The apparent evolutionary maintenance of the KCNL-1 V530L variant in natural *C. elegans* populations seems puzzling given its highly deleterious fitness effects caused by matricidal hatching. Competition experiments indicate, however, that this variant allele can be maintained in more natural conditions mimicking fluctuations in resource availability and/or if reproduction preferentially occurs during early adulthood (*Vigne et al., 2021*). Nevertheless, it remains unknown how constitutively strong egg retention may be beneficial in such conditions, and more generally, how different degrees of *C. elegans* egg retention affect alternative fitness components.

Here, extending our previous work (*Vigne et al., 2021*), we quantified the full spectrum of natural variation in *C. elegans* egg retention using a world-wide panel of strains covering much of the species' genetic diversity (*Lee et al., 2021*). Our first aim was to characterize variation in egg laying, and to identify phylogenetic patterns and genomic regions (QTL) associated with observed phenotypic variation. We show that, while most strains differed only subtly in egg retention, a subset of strains exhibit deviant, strongly increased or reduced, egg retention. We provide evidence for repeated evolution of such extreme egg retention phenotypes through distinct genetic changes. We then characterized a subset of wild strains with divergent egg retention to determine how they differ in egg-laying behaviour and underlying neuromodulatory architecture. These results show that the *C. elegans* egg-laying system harbours surprisingly high natural diversity in the sensitivity to neuromodulators, such as serotonin, suggesting rapid evolution of the involved neural circuitry. In a second objective, we explored why variation in egg retention might be maintained in natural *C. elegans* populations. To address this question, we tested for the presence of fitness costs and benefits associated with variable egg retention of wild *C. elegans* strains. We experimentally demonstrate that strong egg retention usually reduces maternal fertility and survival, mostly due to frequent internal (matricidal) hatching. On the other hand, these genotypes with strong egg retention may be able to benefit from improved offspring protection against environmental insults and from a competitive advantage caused by a significantly reduced extra-uterine egg-to-adult developmental time. Observed natural

variation in *C. elegans* egg-laying behaviour may therefore modify a trade-off between fitness components expressed in mothers versus offspring. Altogether, we present an integrative analysis of natural variation in *C. elegans* egg laying providing first insights into the genetic basis, fitness consequences and possible evolutionary ecological significance of this central reproductive behaviour.

## Results

### Natural variation in *C. elegans* egg retention

Examining a world-wide panel of 316 genetically distinct *C. elegans* wild strains, we found highly variable average egg number in utero (a proxy for egg retention) in self-fertilizing adult hermaphrodites (mid-L4 +48 hr), ranging from approximately 5–50 eggs retained in utero (*Figure 1A–C*). To simplify further analysis of natural variation in egg retention, we defined three phenotypic classes: most strains (81%, N=256) differed relatively subtly in the number of eggs in utero (Class II *canonical* egg retention: 10–25 eggs), including the laboratory reference strain N2. A small number of strains consistently showed either strongly reduced (Class I *weak* egg retention:<10 eggs, 15%, N=46) or increased egg retention (Class III: *strong* egg retention:>25 eggs or larvae in utero, 4%, N=14). Several Class III strains also exhibited internal hatching (*Figure 1—source data 1*).

Strains with extreme egg retention phenotypes at either end of the spectrum were reminiscent of mutants uncovered by screens for egg-laying-defects (*egl* mutants) with major alterations in neural function or neuromodulation (*Trent, 1982*; *Trent et al., 1983*; *Desai and Horvitz, 1989*; *Schafer, 2006*). All examined wild strains showed, however, some egg-laying activity and none of the Class III strains exhibited any obvious, penetrant defects in vulval development or morphogenesis.

To examine if natural variation in egg retention may be linked to strain differences in ecological niche, we tested for effects of geographical and climatic parameters (e.g. latitude/longitude, elevation) (Spearman rank correlations, all p>0.05; data not shown) (*Figure 1—figure supplement 1A*) and substrate type (*Figure 1—figure supplement 1B*) on mean egg retention but found no obvious evidence for such relationships. Strains of all three phenotypic Classes were found across the globe and may co-occur in the same locality, or even in the same microhabitat or substrate (*Figure 1—source data 1*), as previously reported (*Vigne et al., 2021*). A majority (11 out of 14) of Class III strains were isolated in Europe (*Figure 1—figure supplement 1A*), but this may be simply due to increased sampling efforts of this region.

### Both common and rare genetic variants underlie natural differences in egg retention

The presence of significant variation in egg retention allows for mapping of genomic regions that contribute to this variation using GWA mapping. This GWA approach is limited to the detection of common variants as variants below 5% minor allele frequency are excluded from analysis (*Cook et al., 2017*; *Widmayer et al., 2022*). Performing GWA mapping using data for 316 unique strains (isotypes), mapping of mean egg retention yielded a single QTL located on chromosome III spanning 1.48 Mb (*Figure 2A*). This QTL explained 24% of the observed phenotypic variance (*Table 1*). Among the 257 genes found in this QTL region, five have known roles in the egg-laying system (https://www.wormbase.org) and display various natural polymorphisms (*Table 2*). However, we did not detect any obvious candidate polymorphisms in these genes that could explain variable levels of egg retention. Two additional QTL on chromosomes II and IV were detected for variance in egg retention (*CV*) (*Figure 2B*). The three QTLs do not align with any of the recently identified hyper-divergent regions of the genome (*Lee et al., 2021*). The phenotypic distributions and detection of several QTL by GWA indicates that natural variation in *C. elegans* egg retention is likely polygenic, influenced by multiple common variants.

Examining the phylogenetic distribution of extreme egg retention phenotypes based on whole-genome sequence data (*Cook et al., 2017*; *Lee et al., 2021*) indicates that Class I and III phenotypes have been derived multiple times. On average, egg retention did not differ between genetically divergent, likely ancestral, strains (mostly from Hawaii) (N=41) and strains with swept haplotypes (N=275) found across the globe (*Crombie et al., 2019*; *Lee et al., 2021*; *Gilbert et al., 2022*); however, most Class III strains (13 out of 14) exhibited swept haplotypes (*Figure 1—figure supplement 2*). Among the Class III strains with very strong egg retention, four strains (including the newly isolated strain

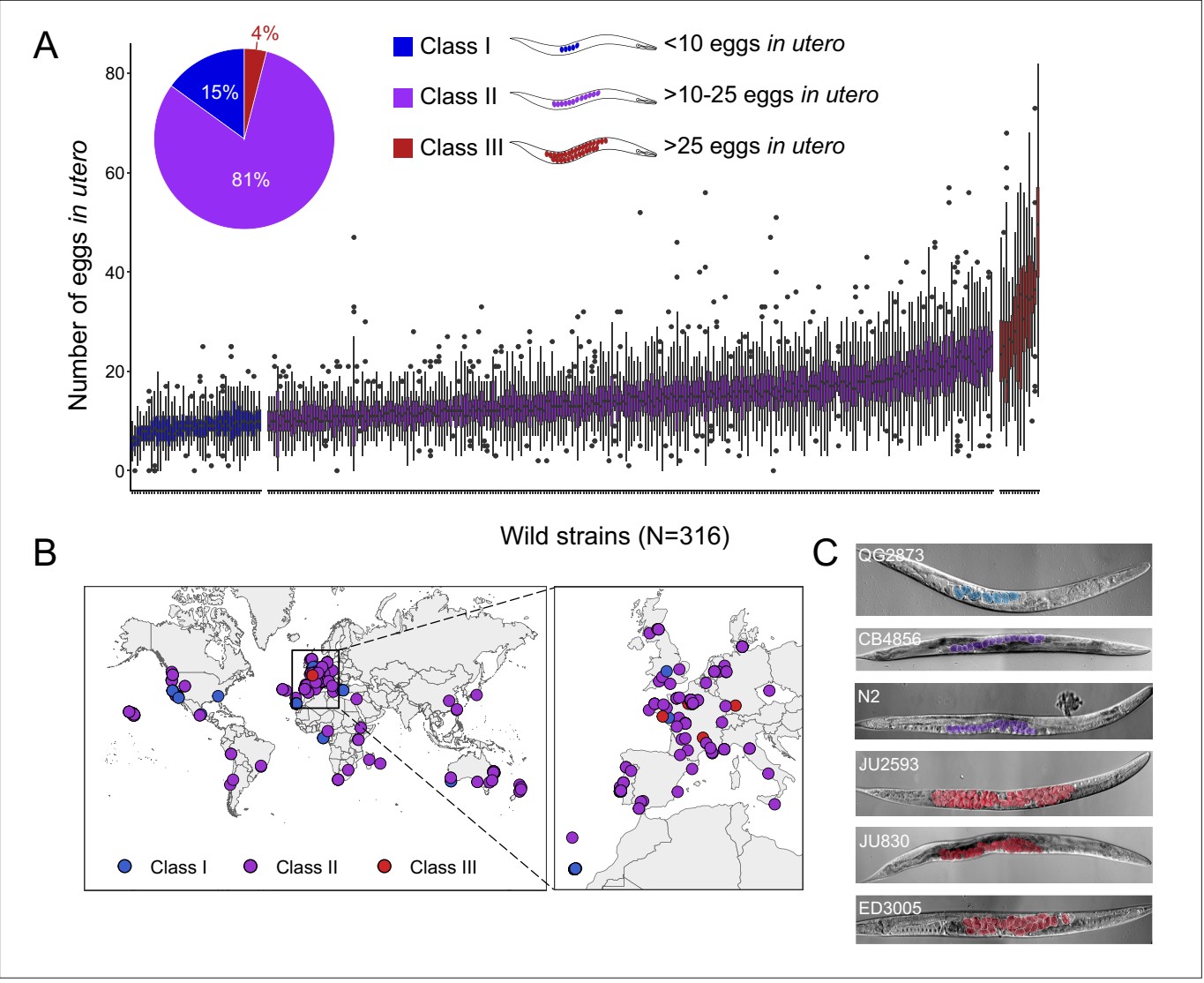

**Figure 1.** Natural variation in *C. elegans* egg retention. (**A**) The number of eggs in utero in hermaphrodites (mid-L4 +48 hr) of 316 genetically distinct strains (isotypes) often strongly deviated from values observed in the laboratory strain N2 (~15 eggs in utero). We defined three classes of strains with distinct levels of egg retention: Class I *weak*:<10 eggs in utero (N=34), Class II *canonical*: 10–25 eggs in utero (N=230), Class III *strong*:>25 eggs in utero (N=14). N=18–150 individuals per strain were scored. (**B**) Geographic distribution of 316 *C. elegans* wild strains. Strains with different degrees of egg retention are labelled in different colours. For a detailed comparison of geographic distribution of the three phenotypic Classes, see ***Figure 1—figure supplement 1A***. (**C**) Nomarski microscopy images of adult hermaphrodites (mid-L4 +48 hr) in wild strains with divergent egg retention. Eggs (coloured) contain embryos at different stages of development.

The online version of this article includes the following source data and figure supplement(s) for figure 1:

**Source data 1.** Excel file containing source data for ***Figure 1***.

**Figure supplement 1.** Natural variation of *C. elegans* egg number in utero.

**Figure supplement 1—source data 1.** Excel file containing source data for ***Figure 1—figure supplement 1***.

**Figure supplement 2.** Egg number in utero in swept versus divergent *C. elegans* strains.

**Figure supplement 2—source data 1.** Excel file containing source data for ***Figure 1—figure supplement 2***.

---

NIC1832) are known to carry a single amino acid substitution (V530L) in KCNL-1 (***Vigne et al., 2021***). This variant has been shown to strongly reduce egg-laying activity (***Vigne et al., 2021***), thus explaining strong egg retention in these four French strains. As previously suggested (***Vigne et al., 2021***), we confirmed that the KCNL-1 V530L is likely derived from a single mutational event, as inferred by local haplotype analysis of the genomic region surrounding *kcnl-1* (***Figure 2C***). In additional Class III strains,

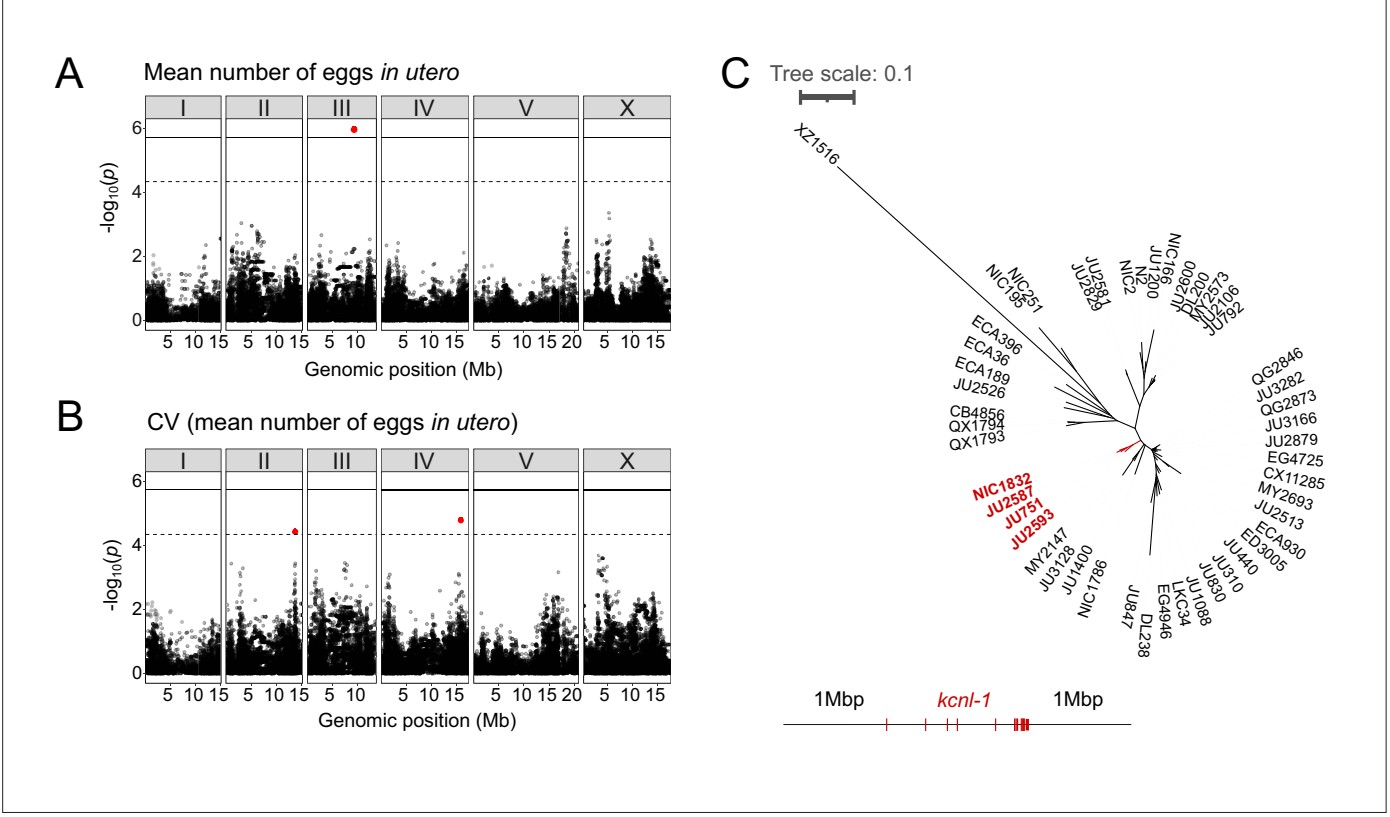

**Figure 2.** Common and rare genetic variants underlie natural differences in egg retention. (**A–B**) Manhattan plots of single-marker based GWA mappings for *C. elegans* egg retention phenotypes (N=316). Each dot represents a SNV that is present in at least 5% of the assayed population. The genomic location of each single-nucleotide variant (SNV) is plotted on the X-axis against its log10(p) value on the y-axis. SNVs that pass the genome-wide EIGEN threshold (dotted line) or the Bonferroni threshold (solid line) are marked in red. (**A**) Manhattan plot of single-marker based GWA mapping region for mean number of eggs in utero. (**B**) Manhattan plot of single-marker-based GWA mapping region for the coefficient of variation (*CV*) (mean number of eggs in utero). (**C**) Neighbour-joining tree based on the 2 Mb region surrounding the *kcnl-1* genomic region using a subset of 48 *C. elegans* wild strains. The four strains with the KCNL-1 V530L variant are shown in red.

The online version of this article includes the following source data for figure 2:

**Source data 1.** Excel file containing source data for *Figure 2*.

which do not carry this variant (N=10), the derived state of constitutively high egg retention must thus be explained by alternative genetic variants. Therefore, natural variation in *C. elegans* egg retention is shaped by both common variants and multiple rare variants, including KCNL-1 V530L, explaining extreme phenotypic divergence.

## Temporal progression of egg retention and internal hatching

To better characterize natural variation in *C. elegans* egg retention, we focused on a subset of 15 strains from divergent phenotypic Classes I-III, with an emphasis on Class III strains exhibiting strong egg retention (at mid-L4 +30 h; *Figure 3A and B*). Class III strains were further distinguished depending on the absence (Class IIIA) or presence (Class IIIB) of the KCNL-1 V530L variant explaining strong egg

**Table 1.** QTL detected by GWA mapping for mean and coefficient of variation (CV) of egg number in utero (N=316 strains).

| Trait | Chromosome | Interval (bp) | Peak | Log10(p) | Variance explained (%) |
|-------|-----------|---------------|------|----------|------------------------|
| Mean | III | 8,532,784–10,019,713 | 9,312,552 | 6.01 | 24.14 |
| CV | II | 13,477,760–14,077,923 | 13,790,719 | 4.45 | 6.74 |
| CV | IV | 15,491,767–16,170,655 | 15,885,539 | 4.81 | 6.74 |

**Table 2.** Potential candidate genes (with known roles in *C. elegans* egg laying) and variants in the QTL interval on chromosome III (GWA mapping for egg number in utero).
Potential high-impact variants are predicted to disrupt gene function, for example, through nonsense or frameshift mutations; low impact variants are predicted to have little or no impact on gene function, such as synonymous mutations (*Cook et al., 2017*).

| Gene | Chromosome | Interval (bp) | Number of variants (predicted high impact) | Number of variants (predicted low impact) |
|------|------------|---------------|--------------------------------------------|-------------------------------------------|
| *pat-2* | III | 8,818,898–8,825,266 | 1 | 2 |
| *lin-12* | III | 9,060,220–9,071,472 | 0 | 2 |
| *ina-1* | III | 9,168,072–9,172,802 | 1 | 2 |
| *lin-52* | III | 9,824,082–9,824,750 | 0 | 1 |
| *cbp-2* | III | 9,923,173–9,924,800 | 20 | 11 |

retention (*Vigne et al., 2021*). In addition, we included the strain QX1430 (Class II), a derivative of the N2 reference strain, in which the major-effect, N2-specific *npr-1* allele, impacting ovulation and egg laying (*Andersen et al., 2014*; *Zhao et al., 2018*), has been replaced by its natural version (*Andersen et al., 2015*); this allowed us to directly assess the effect of the N2 *npr-1* allele on examined phenotypes. Note that this strain selection, especially concerning the largest Class II, is unlikely to reflect the overall strain diversity observed across the species.

The temporal dynamics of egg number in utero varied strongly across the hermaphrodite reproductive span (*Figure 3C*), in line with progeny production and presumptive ovulation rates, coupled to the number of remaining self-sperm (*Ward and Carrel, 1979*; *McCarter et al., 1997*; *Kosinski et al., 2005*; *McMullen et al., 2012*; *Large et al., 2017*; *Zhao et al., 2018*). Towards the end of the reproductive period (between mid-L4 +48 hr to mid-L4 +72 hr), surviving animals in all phenotypic Classes showed strongly reduced offspring numbers in utero (*Figure 3C*). Scoring the age distribution of progeny in utero of young (mid-L4 +30 hr) adult hermaphrodites confirmed that increased egg number in utero was linked to prolonged intra-uterine embryonic development (*Figure 3D*). At this developmental stage, most Class III strains contained more advanced embryonic stages (or L1 larvae) compared to Class I and II strains (*Figure 3D*, *Figure 3—figure supplement 1A*). Examining the temporal dynamics of internal hatching across the entire reproductive span, we found that internal hatching occurred consistently in the six strains with highest egg retention: all Class IIIB strains and in two Class IIIA strains (JU830, JU2829; *Figure 3E*).

Using the 15 focal strains, we further tested if strain differences in egg retention correlate with differences in body or egg size, that is, morphological characteristics likely modulating the capacity to retain eggs. Both egg and body size (of early adults at beginning of fertilization, containing 1–2 eggs in utero) showed significant differences across strains (*Figure 3—figure supplement 1B and C*); in addition, there was a trend for a correlation between body and egg size across strains (*Figure 3—figure supplement 1D*). However, egg and body size did not correlate with measures of mean egg retention across strains (*Figure 3—figure supplement 1E and F*), suggesting that the capacity to retain eggs is not simply a function of body or egg size.

## Natural variation in egg-laying behaviour

Given that egg number in utero is modulated by temporal changes in rates of both ovulation and egg laying (*Figure 3C*), we wanted to determine to what extent natural variation in *C. elegans* egg retention can be attributed to differences in egg-laying behaviour. We first examined the temporal dynamics of egg-laying activity across the hermaphrodite reproductive span by measuring the number of eggs laid during a two-hour window at five distinct adult ages (*Figure 4*). In line with the temporal progression of egg retention (*Figure 3C*), egg-laying activity was highly dynamic, peaking between mid-L4 +24 hr and mid-L4 +30 hr for most strains, followed by a decline until cessation of reproduction (*Figure 4*). Most Class III strains exhibited a shortened time window of egg-laying activity, with few eggs being laid from mid-L4 +48 hr onwards (*Figure 4*).

We next quantified natural variation in behavioural patterns of egg laying during peak activity (mid-L4 +30 hr). Based on continuous video imaging of the reference strain N2, *C. elegans* egg laying

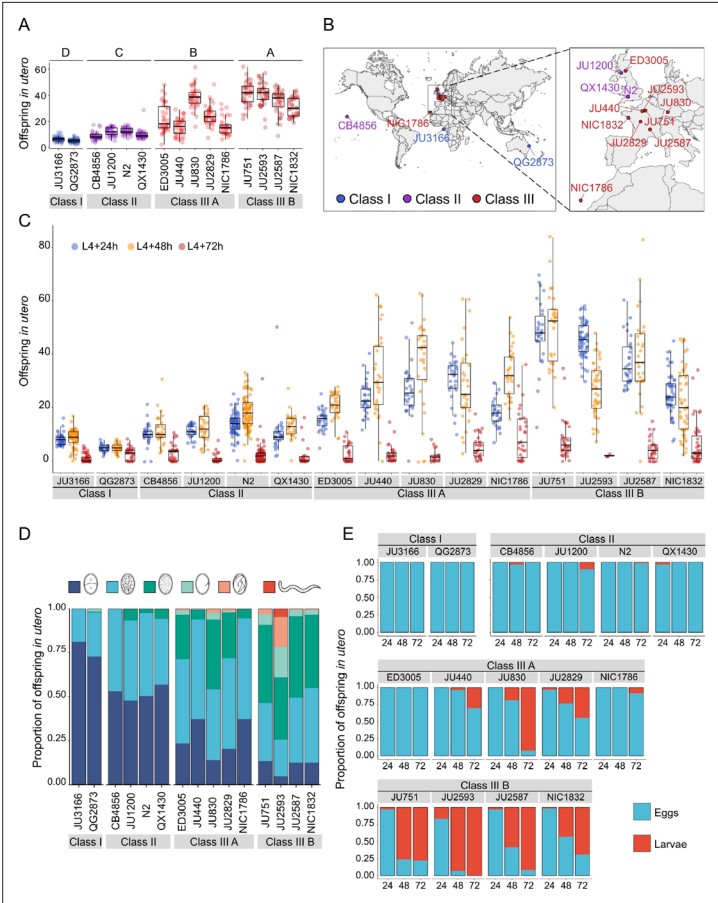

**Figure 3.** Temporal progression of egg retention and internal hatching. (**A**) Egg retention in a subset of 15 strains with divergent egg retention, divided into the three phenotypic classes. Class I *weak*:<10 eggs in utero (N=34), Class II *canonical*: 10–25 eggs in utero (N=230), Class III *strong*:>25 eggs in utero (N=14). Class III strains were further distinguished depending on the absence (Class IIIA) or presence (Class IIIB) of the KCNL-1 V530L variant explaining strong egg retention. Estimates of *Class* effects labelled with the same letter are not significantly different from each other (Tukey's honestly significant difference, p>0.05) based on results of a Two-Way ANOVA, fixed effect *Class:* $F_{3,577}$=710.38, p<0.0001, fixed effect *Strain(nested in Class)*: $F_{11,577}$=33.58, p<0.0001. (**B**) Geographic distribution of the 15 focal strains with divergent egg retention. (**C**) Temporal dynamics of offspring number in utero in the 15 focal strains. Number of eggs and larvae in utero at three stages covering the reproductive span of self-fertilizing hermaphrodites. N=28–96 individuals per strain per time point (except for JU2593: at mid-L4 +72 hr: only four individuals were scored as most animals were dead by this time point). (**D**) Age distribution of embryos retained in utero of hermaphrodites (mid-L4 +30 hr) in the 15 focal strains. Embryonic stages were divided into five age groups according to the following characteristics using Nomarski microscopy (*Hall and Altun, 2007*): 1–2 cell stage, 4–26 cell stage, 44 cell to gastrula stage, bean to two-fold stage, three-fold stage, L1 larva. (Data from same cohort of animals used for experiment shown in **A**). (**E**) Frequency of internal hatching across three time points of the reproductive span of self-fertilizing hermaphrodites (extracted from data shown in **D**). Red bars indicate the proportion of individuals carrying at least one L1 larva in the uterus; blue bars indicate the proportion of individuals carrying only embryos in the uterus. Dead mothers were excluded from analyses.

The online version of this article includes the following source data and figure supplement(s) for figure 3:

**Source data 1.** Excel file containing source data for *Figure 3*, *Figure 3—figure supplement 1A*.

**Figure supplement 1.** Temporal progression of egg retention and internal hatching.

**Figure supplement 1—source data 1.** Excel file containing source data for *Figure 3—figure supplement 1B–F*.

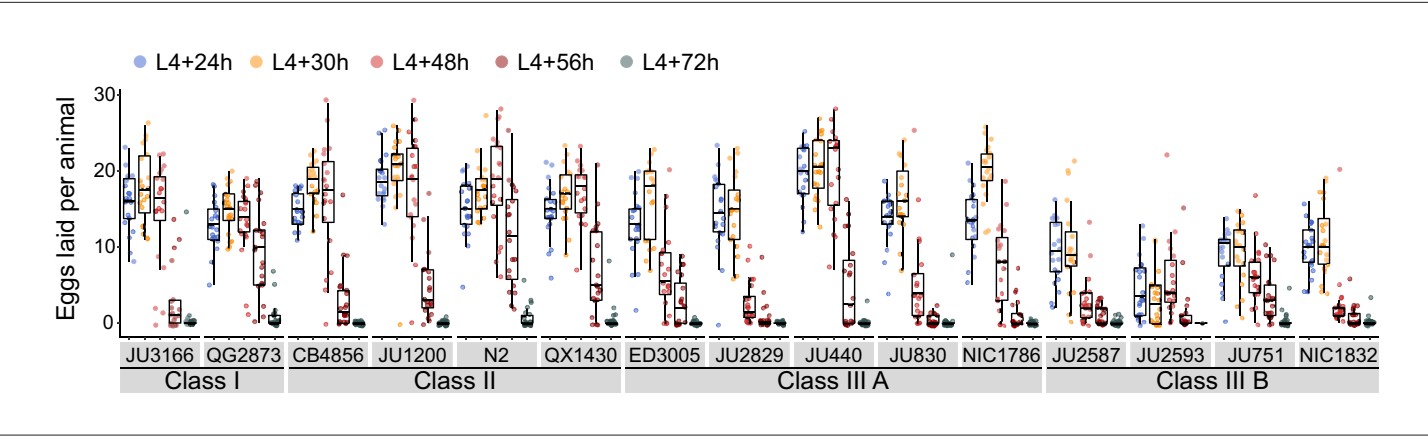

**Figure 4.** Natural variation in egg-laying activity. Temporal dynamics of egg-laying activity in the 15 focal strains. Number of eggs laid (within a two-hour window) at five time points across the reproductive span of self-fertilizing hermaphrodites. N=20 individuals per strain per time point except for JU2593 (at mid-L4 +72 hr: only four individuals could be scored as most animals were dead by this time point). Note that several Class III strains laid eggs containing advanced-stage embryos, evidenced by L1 hatching within the two-hour window of the experiment (*Figure 4—source data 1*).

The online version of this article includes the following source data for figure 4:

**Source data 1.** Excel file containing source data for *Figure 4*.

has been described as a rhythmic two-state behaviour with inactive (~20 min) and active (~2 min) phases, during which multiple eggs are expelled (*Waggoner et al., 1998*). Here, we used a simplified, non-continuous (scan-sampling) method to detect variation in egg-laying behavioural patterns across the 15 focal strains: we scored the presence and number of eggs laid by isolated adults every 5 min over a 3-hr period (*Figure 5A*). This assay enables to estimate egg-laying frequency and to derive an approximate duration of prolonged inactive egg-laying periods, but it does not provide the means to determine the precise timing and structure of active egg-laying periods. Our assay results indicate the presence of significant natural variation in *C. elegans* egg-laying behaviour. The number of intervals with egg laying varied significantly between strains and Classes (*Figure 5B and C*). In line with their egg retention phenotype, Class I strains exhibited an increased number of intervals with egg laying, whereas Class IIIB strains showed a reduced number of such intervals (*Figure 5C*). In contrast, the number of intervals with egg laying did not significantly differ between Class II and IIIA strains (*Figure 5C*). Moreover, strains with strong egg retention tended to exhibit prolonged periods during which egg laying was inactive (*Figure 5D*). In addition, the reference strain N2 (Class II) expelled significantly more eggs per interval (with egg laying) compared to all other strains, including QX1430 – therefore, this atypical phenotype likely caused by the N2-specific *npr-1* allele (*Figure 5E*). Taken together, these observations show that strain and Class differences in *C. elegans* egg-laying behaviour partly align with observed differences in egg retention.

We next wanted to examine how strain differences in egg retention relate to possible differences in the initial onset of egg laying as a function of egg accumulation (*Ravi et al., 2018*). We therefore tested how egg accumulation in utero correlates with the beginning of egg-laying behaviour by measuring measured the timing and onset of the first egg-laying event in the 15 focal strains. Tracking individual hermaphrodites from mid-L4 onwards, we determined the time interval between the time points of first fertilization (presence of one or two eggs in utero) and first egg-laying event (when we also measured egg number in utero). Hermaphrodites of Class II strains laid their first egg after accumulating ~5–10 eggs in utero, that is approximately 1–2 hr after first fertilization, similar to what has been reported previously for the reference strain N2 (*Ravi et al., 2018*; *Figure 6A and B*). By comparison, the time points of the first egg-laying event for Class I and III strains were significantly advanced and delayed, respectively, with corresponding differences in the number of eggs accumulated in utero (*Figure 6A and B*).

The above experiments demonstrate that a substantial portion of the natural variation in *C. elegans* egg retention can be attributed to differences in egg-laying behaviour. First, strain differences in egg retention are established prior to the onset of egg-laying behaviour through a differential delay of the first egg-laying event (*Figure 6A and B*). Hence, the sensitivity to stimuli that trigger the onset of

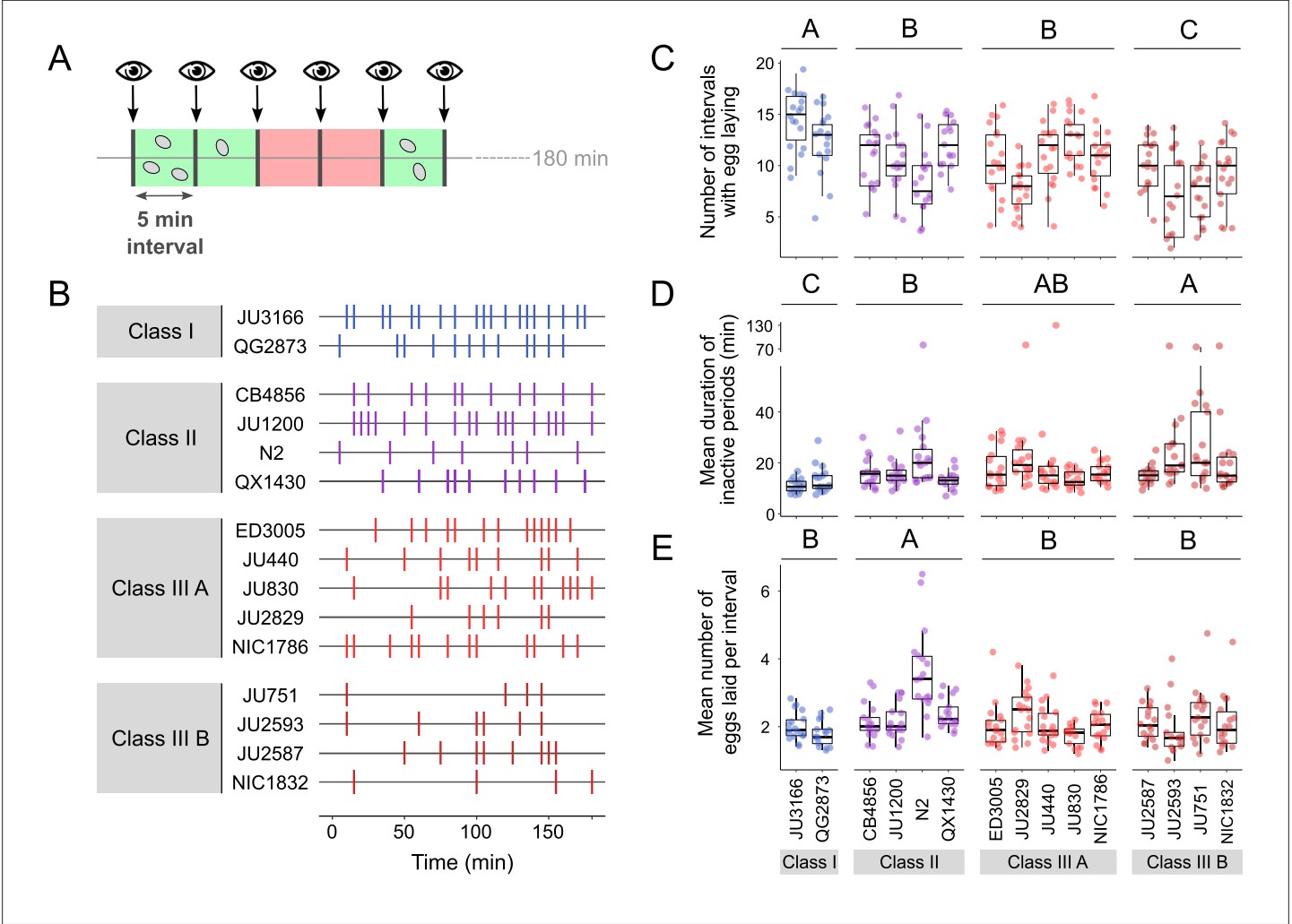

**Figure 5.** Natural variation in egg-laying behaviour. (**A**) Cartoon depicting the design of our scan-sampling experiment to quantify temporal patterns of *C. elegans* egg-laying behaviour in the 15 focal strains (**B–E**). We scored the presence and number of eggs laid by isolated adults (at mid-L4 +30 hr every 5 min across a 3-hr period, resulting in a total of 36 observations [intervals] per individual [N=17–18 individuals per strain]). Intervals with and without egg laying are marked in green and red, respectively. (**B**) Raster plots illustrating strain variation in temporal patterns of egg-laying behaviour across 3 hr of observation. Each horizontal line represents single individual and vertical bars indicate 5-min intervals during which one or more eggs were laid. For a detailed figure of the same data, see *Figure 5—figure supplement 1A*. (**C**) The number of 5-min intervals with egg laying differed significantly between strains and Classes. Two-Way ANOVA, fixed effect *Class*: $F_{3,248}$=19.94, p<0.0001, fixed effect *Strain(nested in Class)*: $F_{11,248}$=5.20, p<0.0001. Estimates of *Class* effects labelled with the same letter are not significantly different from each other (Tukey's honestly significant difference, p<0.05). Each dot represents the number of intervals with egg laying (out of a total of 37 intervals) per individual (N=17–18 individuals per strain). (**D**) The estimated mean duration of inactive periods (min) differed significantly between strains and Classes. Two-Way ANOVA, fixed effect *Class*: $F_{3,248}$=8.46, p<0.0001, fixed effect *Strain(nested in Class)*: $F_{11,248}$=2.93, p=0.0012. Estimates of *Class* effects labelled with the same letter are not significantly different from each other (Tukey's honestly significant difference, p>0.05). For each individual, this value was estimated as the mean time (corresponding to the number of intervals without egg laying) separating successive intervals with egg laying. Each dot represents the mean duration of inactive periods (min) per individual (N=17–18 individuals per strain). (**E**) The mean number of eggs laid per interval with egg laying differed between strains and Classes. Two-Way ANOVA, fixed effect *Class*: $F_{3,248}$=12.41, p<0.0001, fixed effect *Strain(nested in Class)*: $F_{11,248}$=6.30, p<0.0001. Estimates of *Class* effects labelled with the same letter are not significantly different from each other (Tukey's honestly significant difference, p>0.05). These significant effects are exclusively explained by the higher value of the N2 strain (Class II) relative to all other strains (p<0.05); none of the strains other than N2 did differ significantly from each other. Each dot represents the mean number of eggs laid per interval with egg laying per individual (N=17–18 individuals per strain). For data on total number eggs laid during the three-hour experiment, see *Figure 5—figure supplement 1B*.

The online version of this article includes the following source data and figure supplement(s) for figure 5:

**Source data 1.** Excel file containing source data for *Figure 5*.

**Figure supplement 1.** *Natural variation in egg-laying behaviour.*

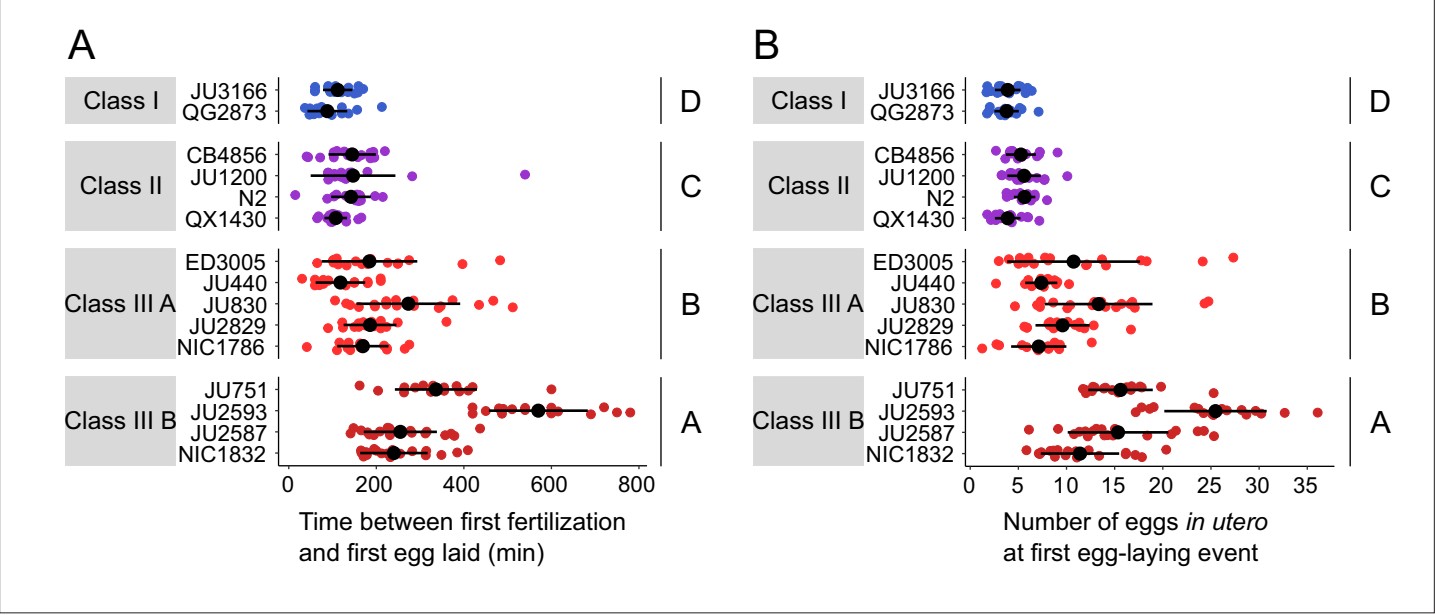

**Figure 6.** Natural variation in egg retention at the first egg-laying event. (**A**) The time (min) between first fertilization and first egg-laying event differed between strains and Classes. Two-Way ANOVA, fixed effect *Class:* $F_{3,261}$=102.38, p<0.0001, fixed effect *Strain(nested in Class)*: $F_{11,261}$=9.40, p<0.0001. Estimates of *Class* effects labelled with the same letter are not significantly different from each other (Tukey's honestly significant difference, p>0.05). N=16–22 individuals per strain. (**B**) The number of eggs in utero at first egg-laying event differed between strains and Classes. Two-Way ANOVA, fixed effect *Class:* $F_{3,261}$=198.73, p<0.0001, fixed effect *Strain(nested in Class)*: $F_{11,261}$=13.00, p<0.0001. Estimates of *Class* effects labelled with the same letter are not significantly different from each other (Tukey's honestly significant difference, p>0.05). N=16–22 individuals per strain; measured in the same individuals as shown in (**A**).

The online version of this article includes the following source data for figure 6:

**Source data 1.** Excel file containing source data for *Figure 6*.

egg-laying behaviour – probably mechanical stretch signals modulated by the accumulation of eggs in utero (*Collins et al., 2016*; *Ravi et al., 2018*; *Ravi et al., 2021*; *Medrano and Collins, 2023*) – seems to vary across different strains. Differential mechanosensory perception of egg accumulation could therefore represent a principal mechanism driving natural variation in egg retention. Second, we observed more subtle strain differences in certain behavioural phenotypes at the peak of egg-laying activity, such as the duration of inactive egg-laying periods, which in part correlated with differences in egg retention (*Figure 5C and D*). Consequently, natural variation in egg retention arises from variation in multiple phenotypes that manifest at distinct time points during adulthood, collectively defining *C. elegans* egg laying.

## Natural variation in *C. elegans* egg-laying in response to neuromodulatory inputs

Observed strain differences in *C. elegans* egg-laying behaviour (*Figure 5*, *Figure 6*) potentially arise through genetic variation in the neuromodulatory architecture of the egg-laying circuit (*Figure 7A*). Multiple neurotransmitters, in particular, serotonin (5-HT), play a central role in synaptic and extra-synaptic neuromodulation of egg laying, with HSN-mediated serotonin bursts triggering egg laying (*Trent et al., 1983*; *Waggoner et al., 1998*; *Shyn et al., 2003*; *Collins et al., 2016*; *Figure 7A*). Exogenous application of serotonin, and other neurotransmitters and neuromodulatory drugs, have been extensively used to study their roles in regulating *C. elegans* egg-laying activity (*Horvitz et al., 1982*; *Trent et al., 1983*; *Schafer, 2006*). We therefore tested if the 15 focal strains differ in their response to exogenous serotonin known to stimulate *C. elegans* egg laying (*Horvitz et al., 1982*; *Schafer, 2006*). We used standard assays, in which animals are reared on NGM agar plates with bacterial food, and then at the start of the egg-laying assay, are transferred to liquid M9 buffer without bacterial food. This liquid treatment inhibits egg laying, but adding serotonin overrides inhibition and stimulates egg laying, as reported for the N2 reference strain (*Horvitz et al., 1982*; *Trent et al., 1983*;

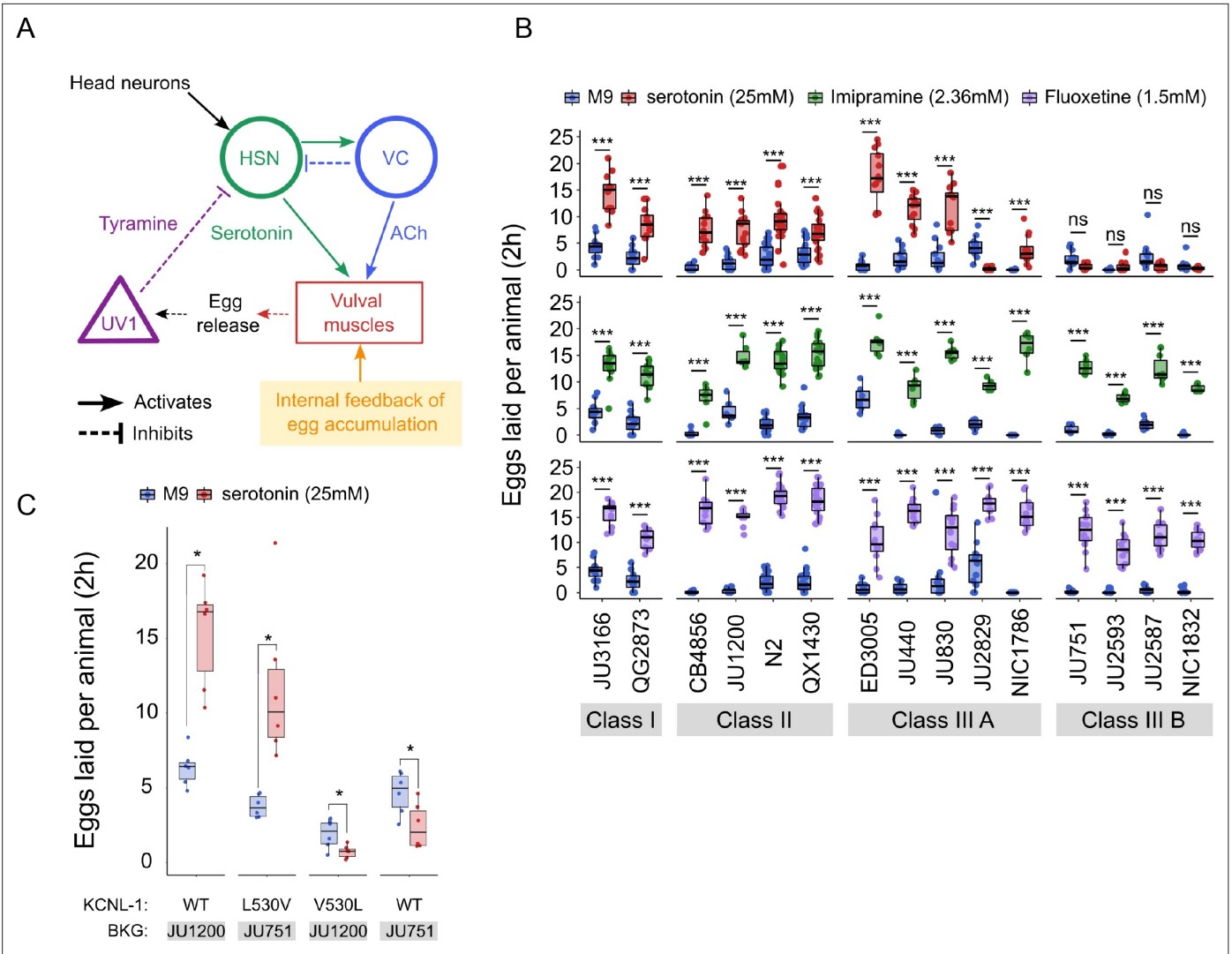

**Figure 7.** Natural variation in *C. elegans* egg-laying in response to neuromodulatory agents. (**A**) Cartoon of the neural circuit controlling *C. elegans* egg laying (*Collins et al., 2016*; *Kopchock et al., 2021*). The structure of the *C. elegans* egg-laying circuit is simple, containing two classes of motoneurons, the two serotonergic hermaphrodite-specific motoneurons (HSN) and the six ventral cord motoneurons (VC), which provide synaptic input to the egg-laying muscles (VM). Serotonin from HSN act through vulval muscle receptors to increase muscle excitability, which together with rhythmic signals from motor neurons, causes contractions of VM during egg laying (*Collins et al., 2016*; *Kopchock et al., 2021*). Mechanical feedback in response to egg accumulation favours exit from the inactive state (*Collins et al., 2016*; *Medrano and Collins, 2023*). Muscles are indicated by rectangles, neurons by circles, and neurosecretory cells (uv1) by triangles. Principal neurotransmitters released by neurons are indicated next to neurons (ACh: Acetylcholine). (**B**) Natural variation in egg-laying activity in response to exogenous serotonin, fluoxetine, and imipramine. Adult hermaphrodites (mid-L4 +30 hr) were placed in M9 buffer without food (control) or in M9 containing the indicated concentrations of serotonin, fluoxetine, and imipramine. The number of eggs laid were scored after two hours. Assays for each of the three treatments were carried out independently; the 15 strains were scored in parallel in both control and treatment conditions for each of the three assays. Serotonin: for each strain, 11–24 replicates (each containing 3.73±0.36 individuals on average) were scored for serotonin and control (M9 buffer) conditions. Align Rank Transform ANOVA, fixed effect *Treatment:* $F_{1,390}=432.62$, p<0.0001, fixed effect *Strain*: $F_{14,390}=42.94$, p<0.0001; interaction *Treatment* x *Strain*: $F_{14,390}=34.40$, p<0.0001. Serotonin stimulated egg-laying in Class, II and III A strains but had no effect on egg laying in Class III B and JU2829 (Class III A) (Tukey's honestly significant difference, ***p<0.0001; ns: not significant). Imipramine: for each strain, 6–18 replicates/wells (each containing 4.62±0.50 individuals on average) were scored for serotonin and control (M9 buffer) conditions. Align Rank Transform ANOVA, fixed effect *Treatment:* $F_{1,222}=562$, p<0.0001, fixed effect *Strain*: $F_{14,222}=23.86$, p<0.0001; interaction *Treatment* x *Strain*: $F_{14,222}=8.52$, p<0.0001. Imipramine stimulated egg laying in strains from the 4 Class (Tukey's honestly significant difference, ***p<0.0001; ns: not significant). Fluoxetine: for each strain, 12–24 replicates/wells (each containing 3.31±0.37 individuals on average) were scored for serotonin and control (M9) conditions. Align Rank Transform ANOVA, fixed effect *Treatment:* $F_{1,378}=1005$, p<0.0001, fixed effect *Strain*: $F_{14,378}=30.09$, p<0.0001; interaction *Treatment* x *Strain*: $F_{14,378}=16.35$, p<0.0001. Imipramine stimulated egg-laying in strains from the 4 Class (Tukey's honestly significant difference, ***p<0.0001; ns: not significant). For detailed statistical results, see *Figure 7—source data 2*. (**C**) Effects of exogenous serotonin (25 mM) on egg laying activity in strains with strongly divergent egg retention due to variation in a single amino acid residue of KCNL-1.

*Figure 7 continued on next page*

*Figure 7 continued*

Strains JU1200$_{WT}$ (canonical egg retention), JU751$_{KCNL-1\ L530V}$ (CRISPR-*Cas9*-engineered, weak egg retention), JU1200$_{KCNL-1\ V530L}$ (CRISPR-*Cas9*-engineered, strong egg retention) and JU751$_{WT}$ (strong egg retention). Adult hermaphrodites (mid-L4 +30 hr) were placed into M9 buffer without food (control) or M9 with serotonin (25 mM). Serotonin stimulated egg-laying in JU751$_{KCNL-1\ L530V}$ and JU1200$_{WT}$ but inhibited egg laying in JU751$_{WT}$ and JU1200$_{KCNL-1\ V530L}$ (Kruskal-Wallis Tests were performed separately for each strain to test for the effect of serotonin on the number of eggs laid; *$p < 0.05$). For each strain, six replicates (each containing 5.50±0.92 individuals on average) were scored for serotonin and control conditions. For additional data, see *Figure 7— figure supplement 1*.

The online version of this article includes the following source data and figure supplement(s) for figure 7:

**Source data 1.** Excel file containing source data for *Figure 7B and C*.

**Source data 2.** Excel file containing statistical results for analyses of data shown in *Figure 7B and C*, *Figure 7—figure supplement 1*.

**Figure supplement 1.** Effects of exogenous serotonin, Imipramine and Fluoxetine on egg laying activity in strains with strongly divergent egg retention due to variation in a single amino acid residue of KCNL-1.

*Weinshenker et al., 1995*). We found exposure to a high dose of serotonin (25 mM) to generate a consistent increase in egg laying in both Class I and II strains (*Figure 7B*). In contrast, responses were highly variable in Class III strains: among the five Class IIIA strains without the *kcnl-1* variant, serotonin stimulated egg laying in all strains except for JU2829, in which serotonin strongly inhibited egg laying as compared to basal egg laying in controls (*Figure 7B*). In addition, serotonin induced a much stronger egg-laying response in the strain ED3005 than in other Class IIIA strains with similar levels of egg retention (*Figure 7B*). ED3005 may thus exhibit serotonin hypersensitivity, which has been observed in certain egg-laying mutants where perturbed synaptic transmission impacts serotonin signalling (*Schafer and Kenyon, 1995*; *Schafer et al., 1996*). In Class IIIB strains carrying the KCNL-1 V530L variant allele, serotonin had no effect on, or a trend towards inhibiting, an already weak egg laying activity (*Figure 7B*). Strains within Classes I, II, and IIIA further showed subtle but significative quantitative differences in the egg-laying response triggered by serotonin despite showing grossly similar egg-laying behaviour in control conditions (*Figure 7B*). Stimulation of egg-laying activity by exogenous serotonin was thus strongly genotype-dependent. We conclude that genetic differences in neuromodulatory architecture of the *C. elegans* egg-laying circuit contribute to natural diversity in egg-laying behaviour.

The action of diverse drugs in clinical use, such as serotonin-reuptake inhibitors (SSRI) and tricyclic antidepressants, have been characterized through genetic analysis of their action on neurotransmitters regulating *C. elegans* egg-laying behaviour (*Trent et al., 1983*; *Weinshenker et al., 1995*; *Weinshenker et al., 1999*; *Ranganathan et al., 2001*; *Dempsey et al., 2005*; *Kullyev et al., 2010*; *Branicky et al., 2014*). As reported for the reference strain N2 (*Dempsey et al., 2005*), we find that the SSRI fluoxetine (Prozac) and the tricyclic antidepressant imipramine strongly stimulated egg-laying activity in all 15 focal strains, which only showed slight (although significant) quantitative differences in drug sensitivity (*Figure 7B*). Both drugs also stimulated egg laying in the Class IIIB strains and the Class IIIA strain JU2829 for which exogenous serotonin either inhibited egg laying or had no effect on it (*Figure 7B*). In the past, mutants unresponsive to serotonin yet responsive to other drugs, including fluoxetine and imipramine, have been interpreted as being defective in the serotonin response of vulval muscles (*Trent et al., 1983*; *Reiner et al., 1995*; *Weinshenker et al., 1995*). This is indeed the likely case of Class IIIB strains carrying the KCNL-1 V530L variant thought to specifically reduce excitability of vulval muscles (*Vigne et al., 2021*). Our results therefore suggest that JU2829 (Class IIIA) may exhibit a similar defect in vulval muscle activation via serotonin caused by an alternative genetic change. Overall, these pharmacological assays do not allow us to conclude if and how HSN function has diverged among strains because the mode of action and targets of tested drugs has not been resolved. Nevertheless, our results are consistent with previous models proposing that these drugs do not simply block serotonin reuptake but can stimulate egg laying, to some extent, through mechanisms independent of serotonergic signalling (*Trent et al., 1983*; *Desai and Horvitz, 1989*; *Reiner et al., 1995*; *Weinshenker et al., 1995*; *Weinshenker et al., 1999*; *Dempsey et al., 2005*; *Kullyev et al., 2010*; *Branicky et al., 2014*; *Yue et al., 2018*).

As shown above, serotonin may also inhibit, rather than stimulate, egg laying, specifically in the Class IIIA strain JU2829, and a similar tendency was observed for Class IIIB strains (*Figure 7B*). This result is in line with past research reporting a dual role of serotonin signalling in *C. elegans* egg laying by generating both excitatory and inhibitory inputs (*Carnell et al., 2005*; *Dempsey et al., 2005*;

*Hobson et al., 2006*; *Hapiak et al., 2009*). In the reference wild type strain (N2), serotonin-mediated inhibitory inputs are masked by its stronger excitatory inputs and are only revealed when function of certain serotonin receptors has been mutationally disrupted (*Carnell et al., 2005*; *Hobson et al., 2006*; *Hapiak et al., 2009*). In analogous fashion, our observations suggest that serotonin-mediated inhibitory effects become only visible in wild strains whose egg-laying response cannot be stimulated by serotonin. In the case of Class IIIB strains, the KCNL-1 V530L variant likely hyperpolarizes vulval muscles to such an extent that exogenous serotonin is insufficient to induce successful egg laying while the negative serotonin-mediated inputs are able to exert their effects, likely through HSN, as previously shown for N2 (*Carnell et al., 2005*; *Hobson et al., 2006*; *Hapiak et al., 2009*). To test if KCNL1- V530L is indeed sufficient to unmask inhibitory inputs of serotonin, we examined the effects of this variant introduced into a Class II genetic background (strain JU1200); the resulting CRISPR-*Cas9*-engineered strain (JU1200$_{KCNL-1\ V530L}$) recapitulates reduced egg laying and strong egg retention (*Vigne et al., 2021*). Very similar to what we found for JU751 (Class IIIB), serotonin had a negative effect on egg laying in JU1200$_{KCNL-1\ V530L}$, suggesting that altering vulval muscle excitability by means of KCNL-1 V530L was indeed sufficient to abrogate positive but not negative serotonin inputs acting in the *C. elegans* egg-laying system (*Figure 7C*). In addition, both fluoxetine and imipramine stimulated egg laying in JU1200$_{KCNL-1\ V530L}$ (*Figure 7—figure supplement 1*), very similar to what we observed in JU751 and other Class IIIB strains (*Figure 7B*). In addition, introducing the canonical KCNL-1 sequence into JU751 (strain JU751$_{KCNL-1\ L530V}$) fully restores egg-laying responses as for the ones observed in JU1200 (*Figure 7C*, *Figure 7—figure supplement 1*). These results thus further confirm that variation in a single amino acid residue of KCNL-1 is sufficient to explain drastic differences in the *C. elegans* egg-laying system and its response to various neuromodulatory inputs.

## CRISPR-*Cas9*-mediated manipulation of endogenous serotonin levels uncovers natural variation in the neuromodulatory architecture of the *C. elegans* egg-laying system

To consolidate the finding that *C. elegans* harbours natural variation in the neuromodulatory architecture of the egg-laying system, we examined how an identical genetic modification – which increases availability of endogenous serotonin levels – affects egg-laying activity in different genetic backgrounds. The *C. elegans* genome encodes a single serotonin transporter (SERT), *mod-5*, which is involved in serotonin re-uptake (*Ranganathan et al., 2001*). We introduced a point mutation corresponding to the *mod-5(n822)* loss-of-function allele, increasing endogenous serotonin signalling (*Ranganathan et al., 2001*; *Dempsey et al., 2005*; *Kullyev et al., 2010*; *Jafari et al., 2011*), into 10 wild strains of the three classes with divergent egg retention using CRISPR-*Cas9* genome editing (*Figure 8A*). Consistent with past reports, we found that *mod-5(n822)* tended to increase basal egg-laying activity in liquid M9 buffer without food. However, the size of this effect was genotype-dependent, and several strains did not show a significantly increased egg-laying response (Class I: JU3166, QG2873, Class IIIA:

**Table 3.** Natural variation in egg laying in response to a gradient of low exogenous serotonin concentrations in wild type and *mod-5(n822)* animals (*Figure 8C*, *Figure 8—figure supplement 1B*). Results for statistical analyses testing for the effects of and interactions between genetic *background*, presence of *mod-5(lf)* and *Treatment* (concentration of serotonin) on egg laying (ANOVA).

| Source | DF | Sum of Squares | F Ratio | p |
|---|---|---|---|---|
| *mod-5* | 1 | 67.45 | 186.42 | <.0001* |
| Background | 7 | 149.21 | 58.91 | <.0001* |
| Treatment | 3 | 120.21 | 110.73 | <.0001* |
| *mod-5* x Background | 7 | 8.94 | 3.53 | 0.0010* |
| *mod-5* x Treatment | 3 | 15.77 | 14.53 | <.0001* |
| Background x Treatment | 21 | 95.77 | 12.60 | <.0001* |
| *mod-5* x Background x Treatment | 21 | 13.31 | 1.75 | 0.021* |
| Error | 510 | 184.55 | | |

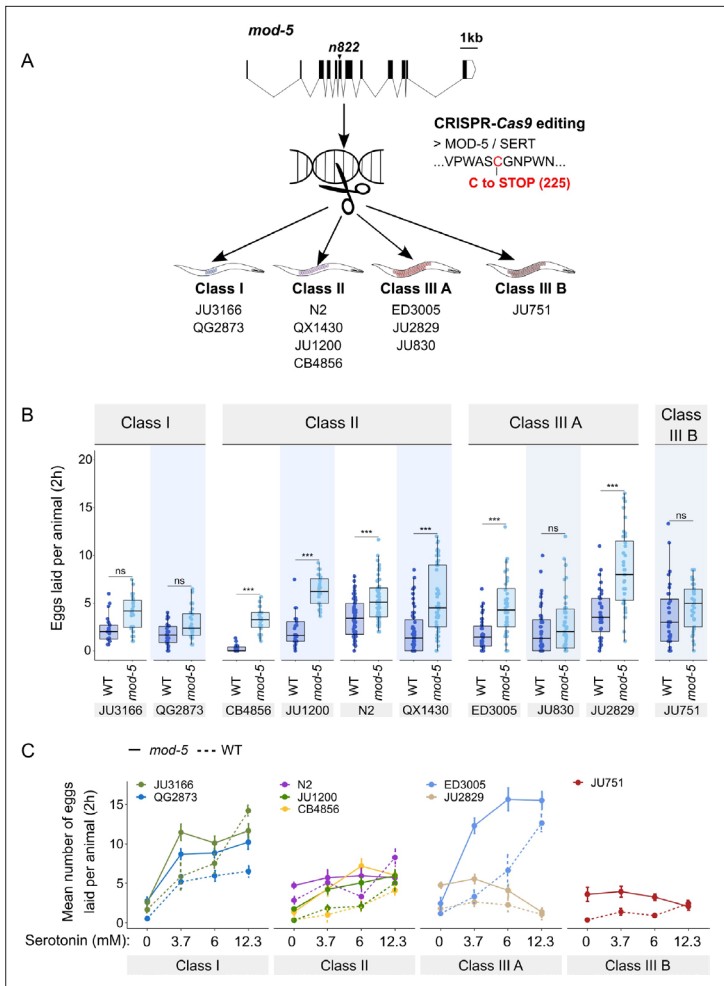

**Figure 8.** Manipulating endogenous serotonin levels uncovers natural variation in the neuromodulatory architecture of the *C. elegans* egg-laying system. (**A**) Cartoon outlining the experimental design to introduce (by CRISPR-*Cas9* technology) the loss-of-function mutation *mod-5(n822)* (*Ranganathan et al., 2001*) into 10 *C. elegans* strains with divergent egg-laying behaviour. (**B**) Natural variation in the effect of *mod-5(lf)* on egg laying (adult hermaphrodites, mid-L4 +30 hr). *mod-5(lf)* increased basal egg-laying activity relative to wild type in the absence of food (M9 buffer), except for the two Class I strains and the Class III strains JU830 and JU751. The stimulatory effect of *mod-5(lf)* on egg laying also varied quantitatively between strains within the same Class, that is, within Class II and Class IIIA. Align Rank Transform ANOVA, fixed effect *mod-5:* $F_{1,709}=175.55$, $p<0.0001$, fixed effect *Background*: $F_{9,709}=20.98$, $p<0.0001$; interaction *mod-5 x Background*: $F_{9,709}=5.34$, $p<0.0001$. N=23–60 replicates per strain, with each replicate containing 3.15±0.18 individuals on average were scored. (**C**) Natural variation in egg laying in response to a gradient of low exogeneous serotonin concentrations in wild type and *mod-5(n822)* animals. Adult hermaphrodites (mid-L4 +30 hr) were placed into M9 buffer without food containing four different concentrations of serotonin (0.0 mM, 3.7 mM, 6.0 mM, 12.3 mM). Strains differed strongly in sensitivity to specific concentrations of exogenous serotonin and these effects of genetic background were further contingent on the presence of *mod-5(lf)* as indicated by the significant three-way interaction term *genetic background* x *mod-5 allele* x *serotonin treatment* (*Table 3*). For complete results of statistical analyses, see *Table 3*; for an alternative representation of the same data, see *Figure 8—figure supplement 1B*. For each concentration, 8–10 replicates (each containing 3.15±0.22 individuals on average) were scored per strain.

The online version of this article includes the following source data and figure supplement(s) for figure 8:

**Source data 1.** Excel file containing source data for *Figure 8*, *Figure 8—figure supplement 1A and B*.

**Source data 2.** Excel file containing statistical results for analyses of data shown in *Figure 8B* and *Figure 8—figure supplement 1A*.

**Figure supplement 1.** Manipulating endogenous serotonin levels uncovers natural variation in the neuromodulatory architecture of the *C. elegans* egg-laying system.

JU830, Class IIIB: JU751) (*Figure 8B*). The effect of *mod-5* was also quantitatively different for certain strains within the same phenotypic Class (Class II and Class IIIA), indicating that serotonin sensitivity of the egg laying system varies genetically despite generating similar behavioural outputs (*Figure 8B*).

Introducing the mutation *mod-5(n822)* did not further increase egg laying when wild strains were exposed to 25 mM serotonin (except for the strain ED3005), likely because egg-laying activity was already at its maximum at such a high dose of serotonin (*Figure 8—figure supplement 1A*). In contrast, exposure to a range of lower serotonin concentrations uncovered significant strain differences in sensitivity to exogenous serotonin and these effects of genetic background were further contingent on the presence of *mod-5(lf)* (*Figure 8C*, *Figure 8—figure supplement 1B*, *Table 3*). Low exogenous serotonin concentrations strongly stimulated egg laying in Class I strains and ED3005 (Class IIIA), and this stimulation was significantly amplified in the presence of *mod-5(lf)* (*Figure 8C*), consistent with the previously observed serotonin hypersensitivity of ED3005 (*Figure 7B*). By contrast, in Class II strains, including N2, low exogenous serotonin only marginally increased egg-laying, and *mod-5(lf)* had little effect (*Figure 8C*). As in previous experiments (*Figure 7B*, *Figure 8B*), we find again that strains sharing the same egg retention phenotype may differ strongly in egg-laying behaviour in response to modulation of both exo- and endogenous serotonin levels (Class IIIA: ED3005 and JU2829) (*Figure 8C*, *Figure 8—figure supplement 1B*).

*C. elegans* wild strains differ in their egg-laying activity in response to the same dose of exogenous serotonin and in response to the same genetic alteration that causes an increase in endogenous serotonin levels. Such significant differences between genotypes in effect sizes of a focal allele are evidence for epistasis (non-linear genetic interactions) (*Gibson and Dworkin, 2004*). Hence, *C. elegans* harbours natural variation in its genetic architecture affecting sensitivity to neuromodulatory inputs of the egg-laying circuit. This genetic variation can occur in elements of the egg-laying circuit, such as the only known natural variant (KCNL-1 V530L) affecting vulval muscle excitability (*Vigne et al., 2021*), but likely also in components acting outside of the core circuit, including head neurons and signalling pathways mediating sensory inputs affecting egg-laying activity (*Aprison et al., 2022*; *Schafer, 2006*).

## Evaluating potential costs and benefits of variation in egg retention

The existence of pronounced natural variation in *C. elegans* egg-laying phenotypes raises the questions of why this variation is maintained and how it might impact alternative fitness components. We

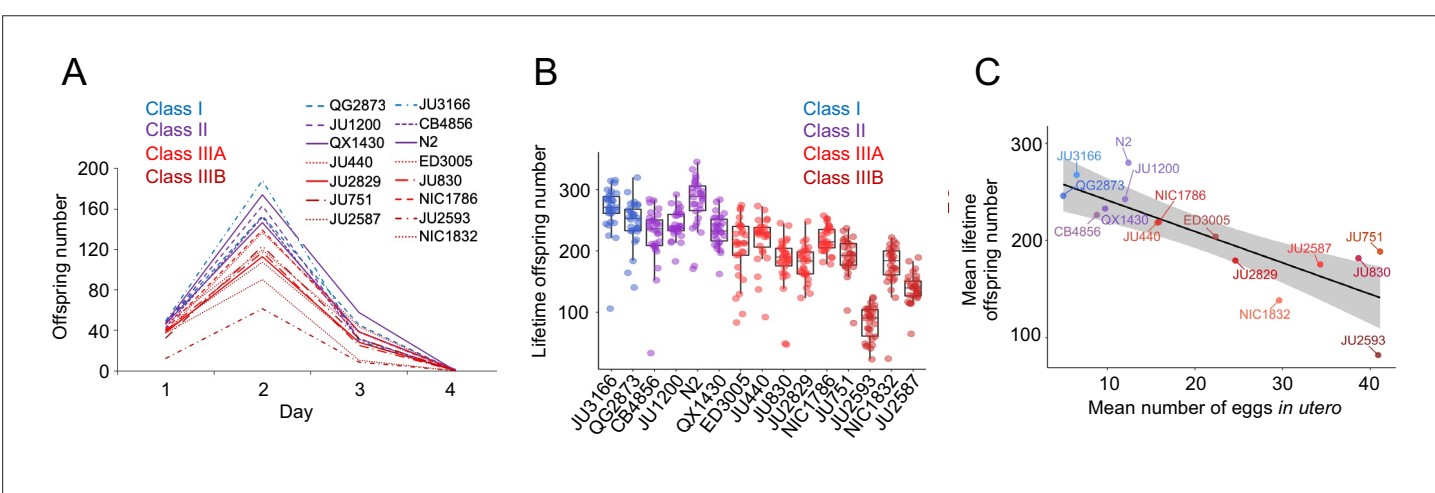

**Figure 9.** Strains with stronger egg retention show reduced lifetime self-fertility and survival. (**A**) Dynamics of lifetime offspring production in self-fertilizing hermaphrodites of the 15 focal strains with divergent egg retention. Hermaphrodites at the mid-L4 stage were isolated to individual NGM plates and their offspring production was scored every 24 hr until reproduction had ceased (~mid-L4+96 hr). N=28–30 individuals per strain. (**B**) Significant differences in total lifetime offspring number between the 15 focal strains (Kruskal-Wallis Test, $\chi^2$=290.79, df=14, p<0.0001). Same experiment as shown in (**A**), N=28–30 individuals per strain. (**C**) Significant negative correlation between mean lifetime offspring number and mean egg retention across the 15 focal strains with divergent egg retention (at mid-L4 +30 hr) ($\rho_{Spearman}$=-0.85, p<0.0001).

The online version of this article includes the following source data for figure 9:

**Source data 1.** Excel file containing source data for *Figure 9*.

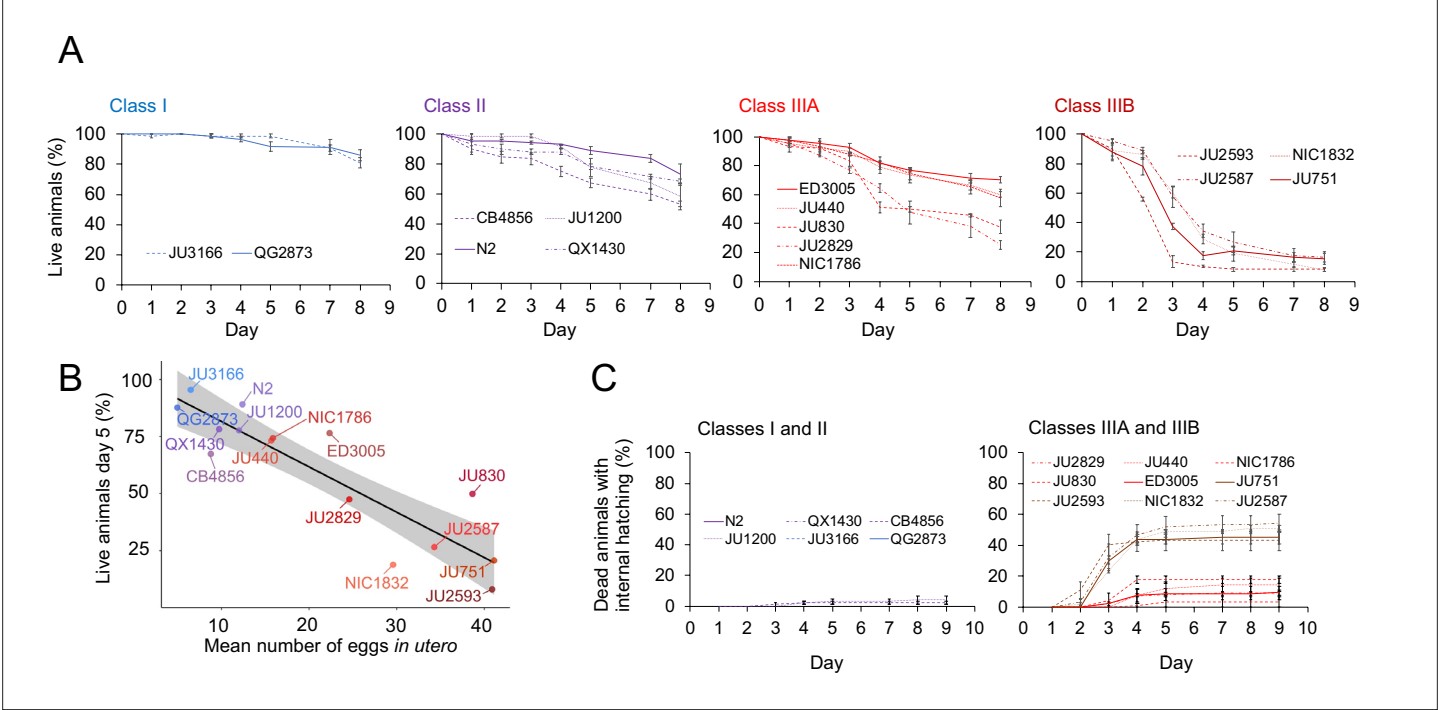

**Figure 10.** Strains with strong egg retention show reduced survival and increased internal hatching. (**A**) Hermaphrodite survival in the 15 focal strains with divergent egg retention, separated by phenotypic classes. Survival was scored every 24 hr across the first eight days of adulthood. For each strain, two to three replicates were scored, with each replicate containing 30–36 individuals. On day 5, the fraction of surviving individuals was significantly different between all four Classes (Tukey's honestly significant difference, all p<0.05) (Two-Way ANOVA, fixed effect *Class*: $F_{3,29}$=178.49, p<0.0001, fixed effect *Strain(nested in Class)*: $F_{11,29}$=6.41, p<0.0001). (**B**) Significant negative correlation between mean percentage of survival (day 5) and mean egg retention across the 15 focal strains with divergent egg retention (at mid-L4 +30 hr) ($\rho_{Spearman}$=-0.84, p=0.00013). (**C**) Temporal progression of internal hatching during the survival assay (from data shown in (**A**)) in the 15 focal strains, measured as the cumulative percentage of dead mothers containing one or more internally hatched larva. For each strain, two to three replicates were scored, and each replicate consisted of 30–36 individuals. No individuals were censored.

The online version of this article includes the following source data for figure 10:

**Source data 1.** Excel file containing source data for *Figure 10*.

therefore experimentally explored potential fitness costs and benefits associated with variable egg retention of *C. elegans* wild strains. We previously reported an apparent fitness cost of strong egg retention but only for a single strain (JU751, Class IIIB), in which mothers die prior to the end of the reproductive span due to internal (matricidal) hatching (*Vigne et al., 2021*). Here, using comparisons between multiple focal strains with variable degrees of egg retention, we tested if increased egg retention consistently generates negative fitness effects. Consistent with this hypothesis, strains with stronger egg retention generally showed reduced lifetime self-fertility and reduced survival, with strongest reductions observed in Class IIIB strains (*Figure 9A-C*). Class III strains showed frequent internal hatching, with mothers dying prematurely, sometimes before the end of the reproductive span (*Figure 10A-C*). Internal hatching in Class I and II strains was absent or occurred only in rare instances at the end of the reproductive period (*Figure 10C*), similar to what has been previously observed for the strain N2 (*Pickett and Kornfeld, 2013*).

To further test when and how strong egg retention and internal hatching may perturb self-fertilization, we screened DAPI-stained hermaphrodites at different adult stages to count remaining self-sperm and internally hatched larvae, and we scored animals for physical damage of the maternal germline and soma. First, a fraction of adults in multiple Class III (but not Class I or II) strains had self-sperm remaining at the time of internal hatching (*Figure 11*), confirming that internal hatching can indeed occur during the reproductive span, potentially disrupting self-fertilization. Consistent with this latter scenario, in the Class IIIB strain JU2593, the total number of self-sperm greatly exceeded the number of lifetime progeny (*Figure 11—figure supplement 1A*). Second, internally hatched larvae in

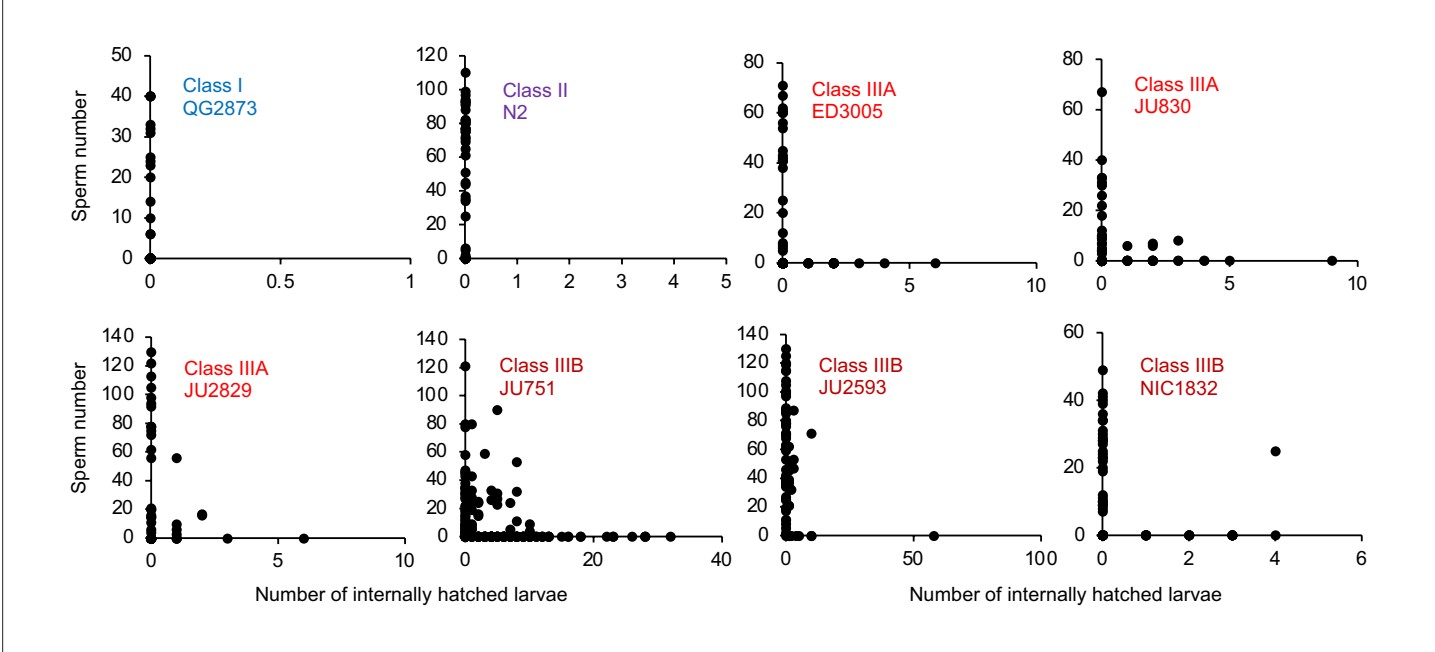

**Figure 11.** Internal hatching may occur during the reproductive span of hermaphrodites. Quantifying hermaphrodite self-sperm numbers in select focal strains with divergent egg retention and testing for the co-occurrence of internally hatched larvae and self-sperm. Adult hermaphrodites derived from age-synchronized populations were collected across multiple time points of their reproductive span, then stained with DAPI to visualize and count spermatids (N=46–175 individuals per strain). Sperm and internally hatched larvae were found to co-occur in multiple Class III strains but not in strains of Class I or II.

The online version of this article includes the following source data and figure supplement(s) for figure 11:

**Source data 1.** Excel file containing source data for *Figure 1*.

**Figure supplement 1.** Internal hatching may have deleterious effects on germline integrity and reproduction.

**Figure supplement 1—source data 1.** Excel file containing source data for *Figure 11—figure supplement 1*.

live mothers with remaining self-sperm (limited to Class III strains) caused apparent physical damage to the maternal gonad and somatic tissues due to larval movement, sometimes disrupting the uterine wall (*Figure 11—figure supplement 1B*). Third, larval movement also appeared to scatter sperm away from the uterine regions adjacent to spermathecae, which likely reduces fertilization efficacy as certain Class III strains had large numbers of unfertilized oocytes in the uterus before self-sperm was depleted (*Figure 11—figure supplement 1C and D*).

The above experiments show that strong egg retention (and internal hatching) correlates with reduced self-fertility (*Figure 9C*) and reduced maternal survival (*Figure 10B*). Class IIIB strains exhibited the most drastic reduction in reproductive output, only using a fraction of available self-sperm. In contrast, in Class IIIA strains (ED3005, JU440, NIC1786) internal hatching occurred mainly after reproduction had ceased (*Figure 10C*), suggesting that strong egg retention has limited detrimental fitness effects in these strains.

## Strong egg retention may provide a competitive advantage in resource-limited environments

Given its frequent deleterious effects on survival and fecundity, we asked if there could be any counteracting beneficial effects associated with increased egg retention. Strong egg retention reflects prolonged intra-uterine embryonic development (*Figure 3D*), which results in laying of eggs holding advanced-stage embryos, as confirmed by scoring the age distribution of embryos contained in eggs laid by young adults of the 15 focal wild strains (*Figure 12A*). Within laid eggs of Class I and II strains, embryos rarely exceeded the 26-cell stage, whilst in Class III, many embryos had started to differentiate, containing hundreds of cells, including late-stage embryos, sometimes close to hatching (*Figure 12A*). In the context of the rapid *C. elegans* life cycle (~80 hr egg-to-adult developmental

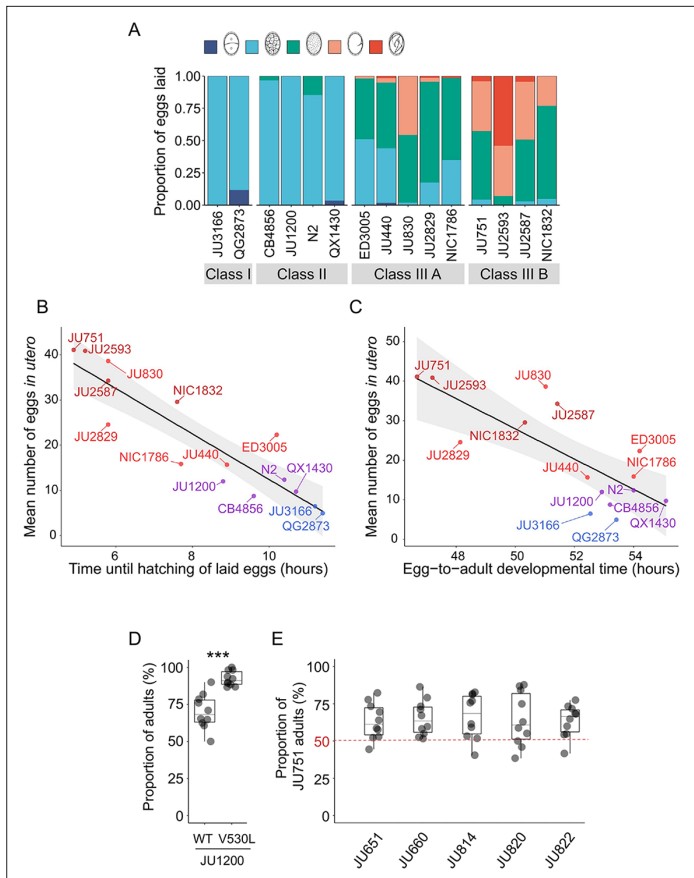

**Figure 12.** Strong egg retention may provide a competitive advantage in resource-limited environments.
(**A**) Age distribution of embryos contained within eggs laid by hermaphrodites (mid-L4 +40 hr) of the 15 focal
strains. Embryonic stages were divided into five age groups according to the following characteristics using
Nomarski microscopy (**Hall and Altun, 2007**): 1–2 cell stage, 4–26 cell stage, 44 cell to gastrula stage, bean to
two-fold stage, three-fold stage, L1 larva. N=45–72 eggs per strain. (**B**) Significant negative correlation between
hatching time of laid eggs and mean egg retention across the 15 focal strains with divergent egg retention
(at mid-L4 +30 hr) ( $\rho_{Spearman}$=-0.92, p<0.0001). Values are estimates of the time point at which 50% of the eggs
had hatched. For each strain, 10–20 adult hermaphrodites (mid-L4 +30 hr) were allowed to lay eggs within an
1-hr window (N=48–177 eggs per strain). The fraction of hatched eggs was scored every hour until all eggs had
hatched. (**C**) Significant negative correlation between egg-to-adult developmental time and mean egg retention
across the 15 focal strains with divergent egg retention (at mid-L4 +30 hr) ( $\rho_{Spearman}$=-0.69, p=0.0041). Values
are estimates of the time point at which 50% of individuals had reached reproductive maturity (one or two eggs
in utero). For each strain, 10–20 adult hermaphrodites (mid-L4 +30 hr) were allowed to lay eggs within a one-
hour window (N=77–318 eggs per strain). After removal of adults, eggs were allowed to hatch and after 45 hr
of development, populations were surveyed every two hours to count the fraction of adults that had reached
reproductive maturity. (**D**) Short-term competition of JU1200$_{WT}$ and JU1200$_{KCNL-1\ V530L}$ against a GFP-tester strain
(*myo-2::gfp*) with a genotype starting frequencies of 50:50. The strain JU1200$_{KCNL-1\ V530L}$ (strong egg retention, Class
III phenotype) outperformed JU1200$_{WT}$ (regular egg retention, Class II phenotype). For each replicate, 20 laid eggs
of either genotype were mixed with 20 laid eggs of the GFP-tester strain on a NGM plate; after 4–5 days (when
food became exhausted) the fraction of GFP-positive adult individuals was determined. Relative to the GFP-tester
strain, JU1200$_{KCNL-1\ V530L}$ showed a significantly higher fraction of adults compared to JU1200$_{WT}$ (Kruskal-Wallis Test,
$\chi^2$=11.57, df=1, p=0.0007). N=10 replicates per genotype. (**E**) Short- term competition of JU751 (Class IIIB, strong
egg retention) against each of five wild strains (Class II, canonical egg retention) isolated from the same locality.
For each of the five strains, 20 freshly laid eggs were mixed with 20 freshly laid eggs from JU751 and allowed to
develop. Adult population size and genotype frequencies were determined after 4–5 days (when food became
exhausted). In each of the five competition experiments, JU751 showed a significantly higher number of adults
(Wilcoxon signed-rank test for matched pairs, all p<0.05). N=10 replicates per strain.

The online version of this article includes the following source data for figure 12:

**Source data 1.** Excel file containing source data for *Figure 12*.

time), observed strain differences in embryonic age are considerable: it takes approximately 2–3 hr to reach the 26-cell stage but around 9–12 hr to develop into late-stage embryos (*Hall and Altun, 2007*). Prolonged egg retention should thus result in earlier hatching ex utero, so that intraspecific differences in the timing of hatching could potentially affect competitive fitness when different genotypes compete for the same resource. Specifically, rapid external hatching may provide an advantage when exploiting resource-limited environments as only a few hours of 'head start' in larval development can be decisive for competitive outcomes, for example, as shown in *Drosophila* flies (*Bakker, 1961*; *Mueller and Bitner, 2015*).

Here, we tested to what extent strain differences in egg retention (of young adults) affect external egg-to-adult developmental time and competitive ability. First, comparing the average timing of larval hatching between the 15 focal strains with variable egg retention, we found that eggs of all Class IIIB strains and some Class IIIA strains hatched on average ~2–5 hours earlier compared to Class I and II strains (*Figure 12B*). Second, to test if earlier larval hatching indeed translates into corresponding earlier onset of reproductive maturity in the 15 focal strains, we measured the time interval between egg laying and reproductive maturity, defined as the onset of fertilization, that is by the presence of 1–2 eggs in utero. Although many Class III strains with strong egg retention reached age at maturity (~2–6 hr) earlier than Class I and II strains, some Class III strains did not maintain their initial head start in larval development (*Figure 12C*). Overall, these measurements confirm that strains with prolonged egg retention benefit from a relatively shorter developmental time from laying to reproductive maturity, which might improve competitive ability. The above experiment only examined eggs laid by young adults (mid-L4 +30 hr); in Class III strains, the observed head start in larval development should thus be further amplified in older hermaphrodites (from mid-L4 +48 hr onwards), containing not only advanced-stage embryos but also larvae as shown earlier (*Figure 3E*).

To quantify the possible effects of egg retention on short-term competitive ability in the absence of confounding genetic variation, we compared the Class II strain JU1200 (wild type) to the engineered JU1200$_{KCNL-1\ V530L}$ strain, i.e. two genetically identical strains with the exception of the engineered KCNL-1 V530L mutation in the latter strain, causing very strong egg retention and laying of advanced-stage embryos: JU1200$_{WT}$ (~15 eggs in utero) versus JU1200$_{KCNL-1\ V530L}$ (~40 eggs in utero; *Vigne et al., 2021*). We competed each strain separately against a green fluorescent protein (GFP)-tester strain by inoculating NGM plates with 20 freshly laid eggs from either genotype. Estimating adult population number and genotype frequencies relative to the GFP-tester strain 5 days after inoculation (around the time point of food exhaustion), the KCNL-1 V530L variant performed significantly better (*Figure 12D*). Given that the principal phenotypic effect of KCNL-1 V530L is reduced egg laying and consequently increased egg retention (*Vigne et al., 2021*), we conclude that strong egg retention can improve competitive ability in food-limited environments.

Finally, to mimic scenarios of ecologically relevant short-term competition between naturally co-occurring wild strains with divergent egg retention, we compared the competitive ability of the Class IIIB strain JU751 (strong egg retention caused by the variant KCNL-1 V530L; ~40 eggs in utero at mid-L4 +30 hr) against strains isolated around the same time from the same habitat (compost heap, Le Perreux-sur-Marne, France) (*Barrière and Félix, 2007*) but exhibiting canonical egg retention (Class II phenotype; hermaphrodites at mid-L4 +30 hr; *Vigne et al., 2021*). For each of five of these strains, 20 freshly laid eggs were mixed with 20 freshly laid eggs from JU751 and allowed to develop. We then estimated adult population size and genotype frequencies 5 days after inoculation (around the time point of food exhaustion). In all five cases, JU751 reached higher adult population sizes at this stage (*Figure 12E*). These experimental results offer preliminary evidence (bearing in mind that our analysis was primarily centered on a single genetic background) that laying of advanced-stage embryos may enhance intraspecific competitive ability, particularly in scenarios where multiple genotypes compete for colonization and exploitation of limited, patchily distributed resources. Similar to our experiments, increased egg-to-adult developmental time has previously been shown to be disadvantageous for *C. elegans* population growth when analysing mutants with increased hermaphrodite sperm production, which incurs a cost as the onset of reproductive maturity will be delayed; thus, although these mutants have the potential to produce overall many more self-progeny, they will be rapidly outcompeted because of delayed maturity whenever resources are limited (*Hodgkin and Barnes, 1991*; *Barker, 1992*; *Cutter, 2004*). Together with theoretical and experimental evidence in insects (*Bakker, 1961*; *Mueller and Bitner, 2015*; *Horváth and Kalinka, 2018*), these observations indicate that maternal

features accelerating offspring development after oviposition can confer fitness benefits in ephemeral habitats with rapidly decaying resources.

## Strong egg retention improves offspring protection when facing sudden environmental stress

Prolonged intra-uterine embryonic development, as observed in viviparous organisms, is thought to provide improved protection of developing embryos against deleterious environmental fluctuations and stressors (*Blackburn, 1999*; *Kalinka, 2015*; *Horváth and Kalinka, 2018*). Whether prolonged egg retention in *C. elegans* can increase such protection is unclear, in particular, because embryos seem already well protected: they are encapsulated within a multi-layered egg shell that is highly resistant to diverse environmental insults, such as osmotic stress or pathogen infections (*Schierenberg and Junkersdorf, 1992*; *Johnston and Dennis, 2012*; *Stein and Golden, 2018*; *Sandhu et al., 2021*). Nevertheless, eggs developing ex utero are still vulnerable to a variety of abiotic and biotic stressors (*Van Voorhies and Ward, 2000*; *Garsin et al., 2001*; *Padilla et al., 2002*; *Burton et al., 2021*; *Fausett et al., 2021*). If eggs in utero are indeed more effectively shielded against environmental insults compared to eggs laid in the external environment, genotypes with strong egg retention may thus benefit from improved offspring protection. Here, we tested this hypothesis by comparing the effects of environmental perturbations on eggs developing in utero versus ex utero using a subset of wild strains with different levels of egg retention.

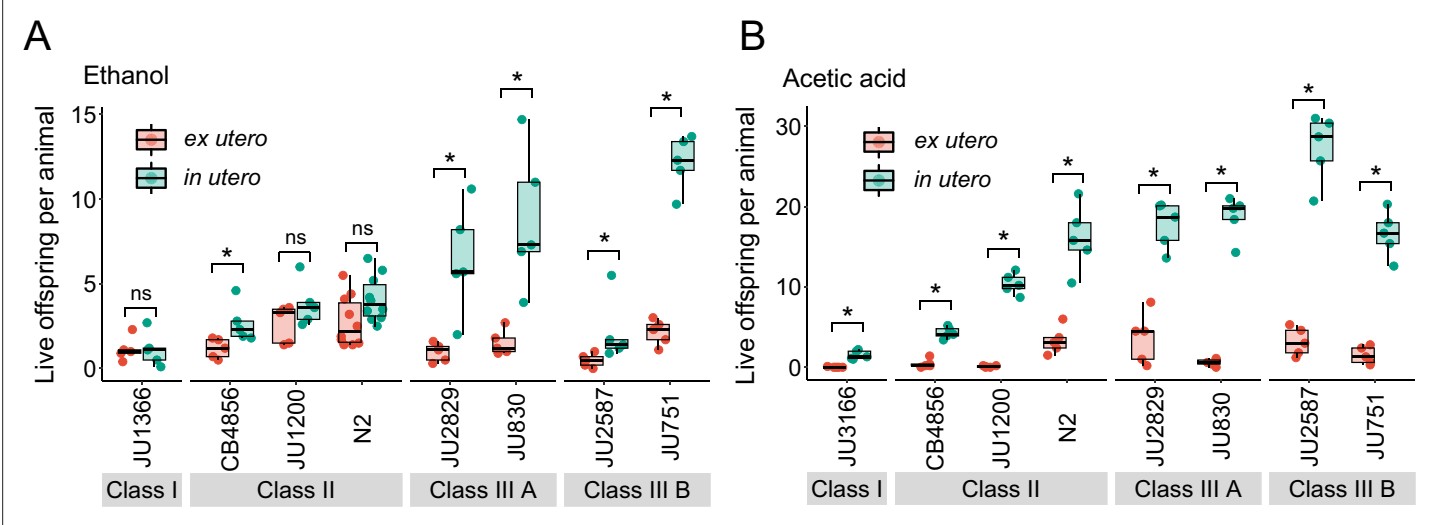

**Figure 13.** Strong egg retention improves progeny protection when facing sudden environmental stress. (**A**) Differences in survival of eggs developing ex utero versus in utero when exposed to a high concentration of ethanol (96%, 10-min exposure). A subset of the 15 focal strains with divergent egg retention was selected to compare the number of surviving eggs ex utero (extracted by dissection) versus in utero (eggs retained in mothers) exposed to ethanol. Overall, the number of surviving eggs tended to be greater when exposed to ethanol in utero compared to ex utero (Kruskal-Wallis Tests performed separately for each strain to compare the number of surviving offspring in utero versus ex utero when exposed to ethanol; *p<0.05, ns: not significant). Class III strains tended to have a higher number of surviving offspring than Class I and II strains. N=5–10 replicates per genotype and treatment (10 hermaphrodites at mid-L4 +30 hr per replicate). See *Figure 13—figure supplement 1A*, for additional (control) data of the experiment. (**B**) Differences in survival of eggs developing ex utero versus in utero when exposed to a high concentration of acetic acid (10 M, 15-min exposure). A subset of the 15 focal strains with divergent egg retention was selected to compare the number of surviving eggs ex utero (extracted by dissection) versus in utero (eggs retained in mothers) exposed to acetic acid. For all strains, the number of surviving eggs was significantly greater when exposed to acetic acid in utero compared to ex utero (Kruskal-Wallis Tests performed separately for each strain to compare the number of surviving offspring in utero versus ex utero when exposed to acetic acid; *p<0.05, ns: not significant). Class III strains tended to have a higher number of surviving offspring than Class I and II strains. N=5 replicates per genotype and treatment (10 hermaphrodites at mid-L4 +30 hr per replicate). See *Figure 13—figure supplement 1B* for additional (control) data of the experiment.

The online version of this article includes the following source data and figure supplement(s) for figure 13:

**Source data 1.** Excel file containing source data for *Figure 13*, *Figure 13—figure supplement 1A and B*.

**Figure supplement 1.** Additional data for experiment shown in *Figure 13A and B*, including data for control conditions (M9 buffer).

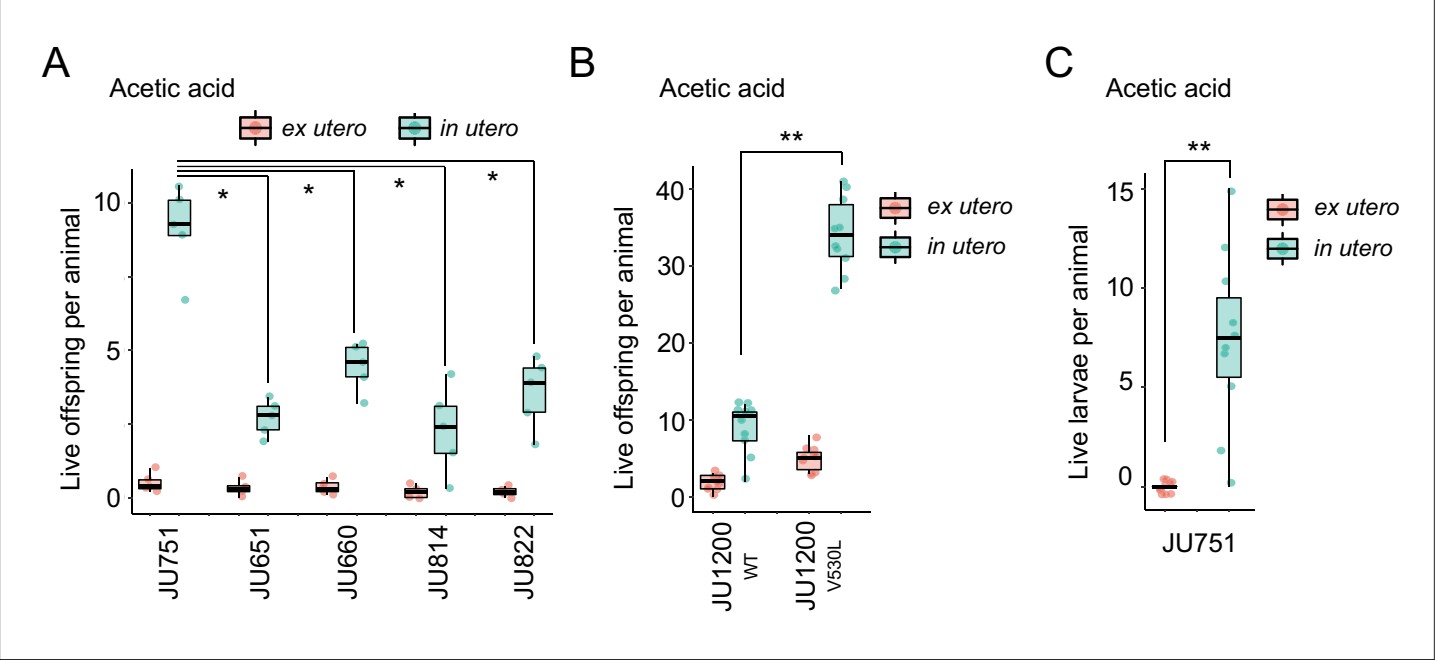

**Figure 14.** Differences in survival of eggs developing ex utero versus in utero when exposed to environmental stress. (**A**) Differences in survival of eggs developing ex utero versus in utero when exposed to acetic acid (10 M, 15-min exposure). Comparison of the strain JU751 (Class IIIB, strong retention) to four strains (Class II, canonical retention) isolated from the same locality. Examining only data for eggs exposed in utero, JU751 exhibited a significantly higher number of surviving offspring compared to all other strains (ANOVA, effect *Strain*: $F_{4,20}$=15.96, p<0.0001; Tukey's honestly significant difference, all p<0.05). N=5 replicates per genotype and treatment (10 hermaphrodites at mid-L4 +30 hr per replicate). See **Figure 14—figure supplement 1** for additional (control) data of the experiment. (**B**) Comparing the Class II strain JU1200$_{WT}$ (canonical retention) and the JU1200$_{KCNL-1}$ $_{V530L}$ strain (strong retention): differences in the number of surviving eggs ex utero (extracted by dissection) versus in utero (eggs retained in mothers) exposed to a high concentration of acetic acid (10 M, 15-min exposure). JU1200$_{KCNL-1 V530L}$ exhibited a significantly higher number of surviving offspring in utero when exposed to acetic acid compared to JU1200$_{WT}$ (Kruskal-Wallis Test, $\chi^2$=14.35, df=1, p=0.0002). N=10 replicates per genotype per treatment. (**C**) Differences in the number of surviving internally hatched larvae exposed to acetic acid (10 M, 15-min exposure) using the strain JU1200$_{KCNL-1 V530L}$: ex utero (extracted by dissection) versus in utero (larvae retained in mothers). The number of live larvae per mother was significantly higher in utero compared to ex utero (Kruskal-Wallis Test, $\chi^2$=13.89, df=1, p=0.0002). N=10 replicates per genotype per treatment.

The online version of this article includes the following source data and figure supplement(s) for figure 14:

**Source data 1.** Excel file containing source data for **Figure 14**, **Figure 14—figure supplement 1**.

**Figure supplement 1.** Additional data for the experiment shown in **Figure 14A**.

We first examined the effects of short-term, acute exposure to high concentrations of ethanol and acetic acid, chemicals that are present in decaying plant matter, that is the natural *C. elegans* habitat (**Félix and Braendle, 2010**). Across strains, most eggs removed from mothers and directly exposed to these chemical treatments were killed, whereas eggs inside mothers exhibited significantly higher survival in most strains, particularly when exposed to acetic acid (**Figure 13A and B**). Eggs in utero were thus partly protected by the body of the mothers, even though mothers instantly died upon treatment exposure. Consequently, increased egg retention resulted in overall higher numbers of surviving offspring per mother in both stress treatments (**Figure 13A and B**).

To corroborate this result, we tested if the Class III strain JU751 displays better maternal protection relative to strains with a canonical egg retention phenotype (Class II) from same habitat, isolated around the same time (**Barrière and Félix, 2007**). Exposure to acetic acid confirmed that higher egg retention of JU751 increased the number of surviving offspring compared to strains with lower retention (**Figure 14A**). In the same fashion, we found the number of surviving offspring in the Class II strain JU1200$_{WT}$ (~15 eggs in utero) to be much reduced compared to JU1200$_{KCNL-1 V530L}$ (~40 eggs in utero) when exposed to acetic acid (**Figure 14B**). In addition, larvae in utero present at the time of treatment were also efficiently protected (**Figure 14C**). Offspring in utero may therefore benefit from maternal protection, so that strains with constitutively higher egg retention will protect overall more progeny when confronting a sudden environmental insult.

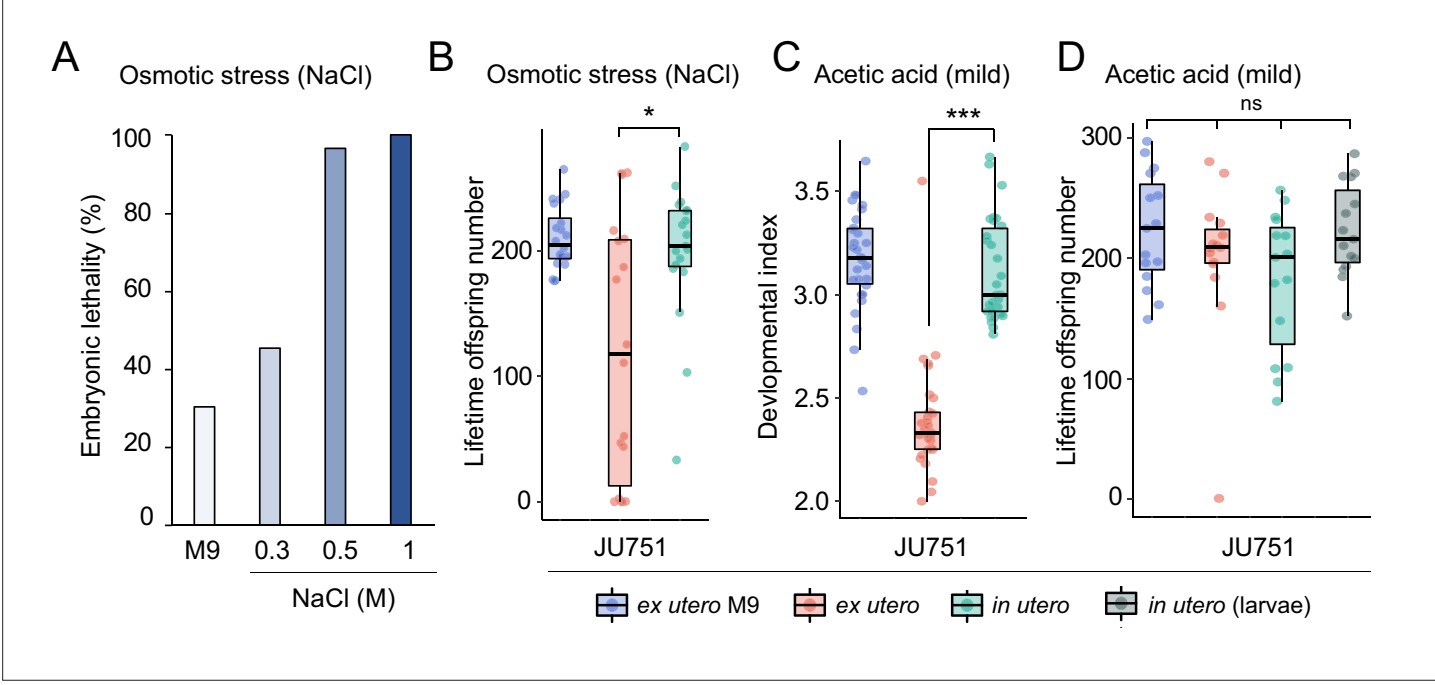

**Figure 15.** Increased egg retention protects offspring viability and fertility under mild environmental stress. (**A**) Effects of osmotic stress on survival of embryos from eggs developing ex utero in the strain JU751. Eggs were extracted from adult hermaphrodites (mid-L4 +36 hr) by dissection and exposed to variable concentrations of NaCl (and M9 control condition) for 15 hr. Embryonic survival was estimated by counting the fraction of live larvae 24 hr later. N=120–288 eggs per treatment. (**B**) Differences in total lifetime offspring production of selfing JU751 animals derived from surviving eggs developing ex utero (dissected from adults at mid-L4 +36 hr) versus in utero when exposed to mild osmotic stress (0.3 M NaCl) for 15 hr; in parallel, ex utero eggs (dissected from adults at mid-L4 +36 hr) were exposed to a control treatment (M9 buffer) for 15 hr. 24 hr after the treatment, larvae from each of the three treatments were allowed to develop and reproduce for 3 days and scored for total offspring production. Animals derived from eggs developing in utero showed significantly higher fertility than animals derived from eggs developing ex utero when exposed to osmotic stress (Kruskal-Wallis Test, $\chi^2$=4.66, df=2, p=0.03). N=18–19 animals per treatment. (**C**) Differences in developmental time of JU751 animals derived from surviving eggs developing ex utero (dissected from adults at mid-L4 +36 hr) versus in utero when exposed to a low concentration of acetic acid (1 M) for 15 min; in parallel, ex utero eggs (dissected from adults at mid-L4 +36 hr) were exposed to a control treatment (M9 buffer). Eggs were then allowed to hatch and develop for 45 hr, at which time we determined their developmental stages. Each stage was assigned a score of development as follows: L3=1; early mid-L4=2; midL4=3; lateL4=4; Adult=5. Each dot represents the mean score reached by offspring produced by one single mother (N=30) mothers per treatment, producing between surviving 9–57 larval offspring. Animals derived from in utero eggs had a significantly higher mean developmental time score, that is they exhibited accelerated development compared to animals derived from *ex utero* eggs (Kruskal-Wallis Test, $\chi^2$=38.93, df=1, p<0.0001). (**D**) Differences in total lifetime offspring production of selfing JU751 animals derived from eggs developing ex utero versus eggs and L1 larvae in utero when exposed to a low concentration of acetic acid (1 M) for 15 min; in parallel, ex utero eggs (dissected from adults at mid-L4 +36 hr) were exposed to a control treatment (M9 buffer). Note that L1 larvae directly exposed to 1 M acetic acid died immediately. Twenty-four hr after the treatment, larvae from each of the four treatments were allowed to develop and reproduce for four days until cessation of reproduction; total offspring production was then scored 24 hr later. There were no significant differences in mean fertility between the four different treatment groups (Kruskal-Wallis Test, $\chi^2$=4.00, df=3, p=0.26). N=15 animals per treatment. Eggs were dissected from adults at mid-L4 +36 hr and L1 larvae at mid-L4 +48 hr.

The online version of this article includes the following source data for figure 15:

**Source data 1.** Excel file containing source data for *Figure 15*.

To test how exposure to milder environmental perturbations affects the reproductive fitness of descendants from eggs exposed in utero *versus* ex utero, we focused on a single strain (JU751, Class IIIB) with constitutively strong egg retention. First, we selected an osmotic stress condition (15-hr exposure to 0.3 M NaCl), which caused relatively low embryonic mortality when eggs were exposed to this treatment ex utero (*Figure 15A*). We then followed surviving individuals to score their lifetime reproductive success under selfing. We found that fertility of individuals derived from stress-exposed eggs ex utero was reduced compared to the ones derived from eggs in utero (*Figure 15B*). Second, we treated eggs in utero versus ex utero with a low concentration of acetic acid (1 M) for 15 min. This treatment was lethal for mothers but did not cause any mortality of embryos laid ex utero. Yet, individuals derived from ex utero eggs developed more slowly and exhibited significantly

delayed reproductive maturity (but not reduced fertility) compared to individuals that hatched from eggs in utero (*Figure 15C and D*). In addition, larvae in utero at the time of stress exposure were also protected by their mother's body and produced as many offspring as animals in control conditions (*Figure 15C*). In contrast, larvae were killed instantly if exposed to this stress ex utero. Together, our experimental results suggest that constitutively strong egg retention can enhance maternal protection of offspring by protecting a relatively larger number of eggs in utero when facing a sudden environmental perturbation. Prolonged egg retention will also reduce the total time of external exposure of the immobile embryonic stage, thus likely lowering risks associated with diverse environmental threats, including exposure to predators and pathogens.

### Natural variation in *Caenorhabditis elegans* egg laying modulates an intergenerational fitness trade-off

Taken together, our experimental results suggest that the degree of *C. elegans* egg retention alters the interplay of antagonistic effects on maternal versus offspring fitness. On the one hand, strong egg retention reduces maternal survival and reproduction (*Figure 9C*, *Figure 10B*); on the other, by prolonging intra-uterine embryonic development, offspring may benefit from improved competitive ability (*Figure 12*) and protection against environmental insults (*Figure 13*, *Figure 14*, *Figure 15*). Therefore, variable egg retention, as observed in natural *C. elegans* populations, may reflect genetic differences in the adjustment of an intergenerational fitness trade-off. In extreme cases of strong constitutive egg retention, *C. elegans* strains exhibit partial viviparity, associated with significantly reduced maternal reproduction and survival. Overall, observed genetically variable duration of intra-uterine development in *C. elegans* thus seems to align well with past reports showing that transitions towards viviparity incur maternal fitness costs as observed for diverse invertebrate and vertebrate taxa with both inter- and intraspecific differences in egg-laying modes (*Avise, 2013*; *Blackburn, 2015*; *Kalinka, 2015*; *Ostrovsky et al., 2016*; *Horváth and Kalinka, 2018*; *Whittington et al., 2022*). To what extent this apparent intergenerational fitness trade-off in *C. elegans* could involve parent-offspring conflict (*Trivers, 1974*) is uncertain. However, given the nearly exclusive selfing mode of reproduction in *C. elegans* (*Barrière and Félix, 2005*; *Lee et al., 2021*), we expect none to very little genetic disparity between hermaphrodite mothers and offspring, indicating much reduced opportunities for intergenerational conflict. Observed natural differences in *C. elegans* egg retention are therefore predicted to reflect properties of genetically distinct (effectively clonally propagating) genotypes, likely resulting from adaptation to distinct ecological niches differing in nature, fluctuations, and predictability of resource availability (*Blackburn, 1999*; *Meier et al., 1999*; *Trexler and DeAngelis, 2003*; *Kalinka, 2015*; *Mueller and Bitner, 2015*; *Dey et al., 2016*; *Horváth and Kalinka, 2018*).

## Discussion

Our study has quantified and characterized natural variation in *C. elegans* egg retention and egg-laying behaviour, resulting in the following main findings: (1) *C. elegans* exhibits quantitative variation in egg retention, with rare instances of extreme phenotypes at either end of the spectrum. (2) Both common and rare genetic variants underlie observed phenotypic variation. (3) Strain differences in egg retention can be largely explained by differences in egg-laying behaviour, such as the onset of egg laying activity and timing of the rhythm between active and inactive egg-laying phases. (4) These behavioural differences map, at least partly, to genetic differences in the sensitivity to various neuromodulators, indicative of natural variation in the *C. elegans* egg-laying circuitry. (5) Modified egg-laying behaviour causing strong egg retention – linked to frequent larval hatching in utero – negatively impacts maternal fitness, but (6) prolonged intra-uterine embryonic development may benefit offspring by improving their competitive ability and protection against environmental insults. (7) Hence, variation in *C. elegans* egg retention may reflect variation in a trade-off between antagonistic effects acting on maternal versus offspring fitness.

*C. elegans* egg retention is a quantitative (complex) trait: observed phenotypic distribution and GWA mapping indicate that this trait is likely influenced by multiple loci of small effect (*Mackay et al., 2009*). In addition, large-effect variants, such as KCNL-1 V530L (*Vigne et al., 2021*), occur but they seem to be relatively rare. Although variation in reproductive processes, such as self-sperm production and ovulation rates, contributes to observed variation in egg number in utero, we also

present clear evidence for natural variation in egg-laying behaviour and involved neuromodulatory processes. How such behavioural differences arise through modification of specific cellular and physiological processes and how these, in turn, are mediated by specific molecular alterations of genes in the *C. elegans* egg-laying system remains to be elucidated. The genetics of the *C. elegans* egg-laying circuit has been studied for decades, and dozens of genes have been identified by mutations, which either reduce or increase egg-laying activity (*Trent et al., 1983*; *Schafer, 2006*). Some of the identified genes encode ion channels regulating cell and synaptic electrical excitability, others encode components of G-protein signalling pathways that act in specific cells of the circuit. Many more genes transmitting environmental and physiological signals via sensory processing are known to regulate *C. elegans* egg laying (*Schafer, 2006*; *Ringstad and Horvitz, 2008*; *Fenk and de Bono, 2015*; *Banerjee et al., 2017*). Recent research has further uncovered an important role of muscle-directed mechanisms regulating egg laying through sensation of internal stretch mediated by egg accumulation in utero (*Collins et al., 2016*; *Ravi et al., 2018*; *Ravi et al., 2021*; *Medrano and Collins, 2023*). This stretch-dependent homeostat directly targets vulval muscles and is sufficient to trigger egg laying when synaptic transmission is reduced or defective, for example, in the absence of HSN command neurons (*Ravi et al., 2018*; *Ravi et al., 2021*; *Medrano and Collins, 2023*). As indicated by our results (*Figure 6B*), changes in such muscle-directed signalling components in response to egg accumulation are likely key factors driving the evolution of *C. elegans* egg retention. Future experiments should therefore be aimed at understanding how such modifications in mechanosensory feedback interact with other potential changes in synaptic transmission to explain the natural diversity in the *C. elegans* egg-laying neural circuit. In addition, we cannot exclude that natural variation in egg-laying behaviour may arise through differential activity of developmental genes generating variation in the neuroanatomy of the egg-laying circuit itself, for example as observed between different *Caenorhabditis* species (*Loer and Rivard, 2007*).

Currently, the only known natural variant modulating *C. elegans* egg laying is the major-effect variant KCNL-1 V530L, which reduces egg laying through likely hyperpolarization of vulval muscles (*Vigne et al., 2021*), so that stimuli, such as mechanosensory inputs, are insufficient to trigger regular and successful egg-laying events. Our new results show that other unknown variants must cause the strong reduction of egg-laying activity in Class IIIA strains. Although our experimental results do not hint at any variants in specific candidate genes or processes, we detected strong differences in the serotonin response between Class IIIA strains (*Figure 7B*). Exogenous serotonin stimulated egg laying in all Class IIIA strains except in JU2829, where it inhibited egg laying as observed for Class IIIB strains (*Figure 7B*). This suggests that JU2829 might carry a variant with effects like KCNL-1 V530L, whereas variants with distinct functional effects would explain reduced egg-laying activity in other Class IIIA strains. The detection of both strong and subtle strain variation in egg-laying activity (also among strains within the same Class), as revealed by our pharmacological assays and when analysing the effects of *mod-5(lf)* in different strains (*Figure 8*), indicates the presence of many different natural variants that modulate egg-laying circuitry in *C. elegans.* Molecular identification of specific variants will thus be essential to explore how the nematode egg-laying circuit evolves at the intraspecific level.

Consistent with previous reports on the apparent costs of transitioning from oviparity to obligate and facultative viviparity (*Avise, 2013*; *Kalinka, 2015*; *Horváth and Kalinka, 2018*), we found that increased *C. elegans* egg retention lowers maternal survival and fecundity. We then show that fitness costs associated with strong egg retention are potentially offset by improved offspring competitive ability and protection against environmental perturbations. To what extent our highly simplified experimental conditions recapitulate relevant ecological scenarios encountered by natural *C. elegans* populations remains to be explored. The likely ecological factors shaping natural variation in *C. elegans* egg retention are thus unknown and we have not found any obvious links between the degree of egg retention and habitat parameters, such as substrate type or climate. However, the likely ancestral (divergent) strains only very rarely exhibited a Class III phenotype, which could indicate that derived state of strong egg retention may have been favoured by the exploitation of novel microhabitats during the historically recent expansion of *C. elegans* reflected by globally distributed, swept haplotypes (*Lee et al., 2019*; *Lee et al., 2021*).

The typical *C. elegans* habitat is highly ephemeral in nature, exhibiting extreme fluctuations in nutrient availability and occurrence of diverse abiotic and biotic variables, including stressors and pathogens (*Frézal and Félix, 2015*; *Schulenburg and Félix, 2017*). A critical omission in our study

is therefore the analysis of plasticity in egg retention and egg-laying behaviour in different or fluctuating environments. *C. elegans* egg laying is known to be highly plastic, strongly modulated by subtle and very diverse environmental factors, so that our analyses in standard laboratory conditions can only capture fractions of the complexity. Most prominently, diverse environmental stimuli (food quantity and quality, osmotic or hypoxic stress, pathogens, etc.) have inhibitory effects on egg laying and long-term exposure to these stimuli will lead to strong egg retention and matricidal hatching (*Trent, 1982*; *Aballay et al., 2000*; *Waggoner et al., 2000*; *Chen and Caswell-Chen, 2003*; *Schafer, 2005*; *Zhang et al., 2008*; *McMullen et al., 2012*; *Fenk and de Bono, 2015*; *Vigne et al., 2021*). Future research will therefore require to carefully examine how such environmental effects modulate egg laying and fitness consequences in strains with the here reported differences in constitutive egg retention. In particular, plastically induced egg retention and matricidal hatching by prolonged starvation may allow for nutrient provisioning of internally hatched larvae as they can consume debris stemming from the decaying mother, hence allowing for developmental growth in the absence of external food (*Chen and Caswell-Chen, 2003*; *Chen and Caswell-Chen, 2004*). Future experiments should therefore specifically test if internally hatched larvae in Class III strains (*Figure 3E*) may gain a further advantage through maternal provisioning while developing in utero, specifically when exposed to long-term conditions inhibiting egg laying, such as starvation.

The existence of pronounced natural variation in *C. elegans* egg retention and egg-laying behaviour generates novel opportunities to dissect the molecular genetic basis of intraspecific variation in egg retention, providing an entry point to understand the proximate mechanisms underlying evolutionary transitions between invertebrate ovi- and viviparity. In contrast to vertebrate taxa with intraspecific variation in egg-laying mode, such as certain lizards (*Recknagel et al., 2021*; *Whittington et al., 2022*), the *C. elegans* model allows for rapid and powerful genetic analysis, for example through linkage mapping of natural variants and subsequent functional validation of variants by CRISPR-Cas9 gene editing in multiple strains (*Evans et al., 2021*). Identifying the precise changes in *C. elegans* egg-laying circuitry that have led to variation in egg retention may also generate insights into the genetic basis of evolutionary diversification of egg-laying modes across species and genera. Nematodes display frequent transitions from oviparity to obligate viviparity in many distinct genera (*Sudhaus, 1976*; *Ostrovsky et al., 2016*), including in the genus *Caenorhabditis*, with at least one viviparous species, *C. vivipara* (*Stevens et al., 2019*). Although evidence exists for the evolution of egg-laying circuitry across oviparous *Caenorhabditis* species (*Loer and Rivard, 2007*), the specific cellular and genetic changes responsible for the transition to obligate viviparity in *C. vivipara* have yet to be examined. Resolving the genetic basis of intraspecific variation in *C. elegans* egg retention, including partial or facultative viviparity, may thus shed light on the molecular changes underlying the initial steps of evolutionary transitions from oviparity to obligate viviparity in invertebrates.

## Materials and methods

### Materials availability statement

Requests for additional information, data and strains generated in this study (*Supplementary file 1*) should be directed to the lead contact, Christian Braendle (braendle@unice.fr).

### *C. elegans* strains and culture conditions

A complete list of strains used in this study is provided in *Supplementary file 1*. *C. elegans* stocks were maintained on 2.5% agar NGM (Nematode Growth Medium) plates (55 mm diameter) seeded with *E. coli* strain OP50 at 15 °C or 20 °C (*Stiernagle, 2006*). All strains were decontaminated by hypochlorite treatment (*Stiernagle, 2006*) in the third generation after thawing and kept in ad libitum food conditions on NGM plates. Detailed information for *C. elegans* wild strains used in this study is available at the *Cae*NDR website (https://caendr.org/) (*Crombie et al., 2023*). Most experiments also included the strain QX1430, a derivative of the N2 reference strain, in which the major-effect and N2-specific *npr-1* allele has been replaced by its natural version (*Andersen et al., 2015*). For experiments, all strains were maintained on 2.5% agar NGM (Nematode Growth Medium) plates (55 mm diameter) at 20 °C unless noted otherwise. All data are reported for exclusively selfing (self-fertilizing) hermaphrodites. Liquid assays were performed in 0.1 ml M9 buffer in 96-well microplates (Greiner

Bio-One, Ref: 655180); each well was treated as an independent replicate, containing approximately three worms on average. During the incubation period, plates were mildly agitated on a shaker.

## Method details

### Age-synchronization and identification of developmental stages

Mixed-age hermaphrodite stock cultures were hypochlorite-treated to obtain age-synchronized, arrested L1 larval populations (*Stiernagle, 2006*). Hermaphrodites were then picked at the mid-L4 stage based on the morphology of the vulval invagination (*Mok et al., 2015*).

### Quantification of progeny number in utero and determination of embryonic stages

Progeny number (eggs and larvae) in utero was measured in age-synchronized hermaphrodites, mounted directly on microscopy slides and gently squashed with a coverslip. The number of offspring was then counted using a 20 x DIC microscope objective. An individual was considered to exhibit internal hatching when one or more larvae were visible in the uterus. Embryonic stages in utero were determined using DIC microscopy, and we distinguished six stages: 2 cell stage, 3–20 cell stage, early gastrula stage (21+ cells), comma stage, twofold stage, threefold stage, and L1 larva. To determine the age distribution of embryos contained in laid eggs (*Figure 12A*), age-synchronized hermaphrodite populations (mid-L4 +30 hr) were allowed to lay eggs for 30 min, after which embryonic stages were determined within 20 min.

### Egg size measurements (*Figure 3—figure supplement 1*)

Laid eggs of age-synchronized hermaphrodites (mid-L4 +30 hr) were collected in M9 buffer and mounted on 4% agar pads on glass slides. Microscopy images were acquired with a 40 x DIC microscope objective using Fiji (*Schindelin et al., 2012*). Length and width of eggs were measured to calculate egg volume with the ellipsoid volume function: $4/3 \times \pi \times (length/2) \times (width/2)^2$ (*Fausett et al., 2021*).

### Body size measurements (*Figure 3—figure supplement 1*)

Hermaphrodite body size was measured at the early adult stage prior to the accumulation of eggs in utero, i.e., when they contained 1 or 2 eggs in utero. Animals were collected in M9 and mounted on 4% agar pads on glass slides and microscopy images (10 X objective) were acquired and processed using Fiji (*Schindelin et al., 2012*). As an estimate of body size, we measured the length of each animal from the tip of the head to the end of the tail using the segmented line tool and width was measured at the site of the vulval opening. The volume of the animal was calculated as that of a cylinder: $\pi \times (width/2)^2 \times length$ (*Vielle et al., 2016*).

### Quantification of egg-laying behaviour (*Figure 5*)

Unlike the reference behavioural protocol based on continuous video imaging (*Waggoner et al., 1998*), we used a non-continuous, scan-sampling method (*Vigne et al., 2021*) to estimate variation in egg-laying behavioural patterns across the 15 focal strains. Individual animals at peak activity of egg laying (mid-L4 +30 hr) were isolated onto seeded NGM plates. Using a dissecting microscope, we then scored the presence and number of eggs laid by isolated adults every 5 min over a 3-hr period. We therefore screened a total of 36 successive 5-min intervals (N=17–18 individuals per strain). This assay allows for the estimation of egg-laying frequency and the derivation of an approximate duration for extended inactive egg-laying periods. However, this assay does not have sufficient resolution to determine the exact timing and structure of active egg-laying periods.

### Serotonin, imipramine, and fluoxetine assays (*Figures 6 and 7*)

Experiments testing for the effects of serotonin (5-hydroxytryptamine creatinine sulfate monohydrate, Sigma, Ref: H7752), fluoxetine (Fluoxetine hydrochloride, Sigma, Ref: PHR1394) and imipramine (imipramine hydrochloride, Sigma, Ref: I0899) on egg laying were based on previously established liquid culture protocols and drug concentrations (*Trent et al., 1983*; *Weinshenker et al., 1995*; *Weinshenker et al., 1999*; *Ranganathan et al., 2001*; *Dempsey et al., 2005*; *Kullyev et al., 2010*;

*Branicky et al., 2014*). Approximately three age-synchronized adults (mid-L4 +30 hr) were transferred to individual wells of a 96-well microplate containing serotonin or drugs dissolved in 100 μl of M9 buffer (without bacterial food); in parallel, control animals were transferred to wells containing only M9 buffer. The number of eggs released was scored after 2 hr at 20 °C.

## Introduction of *mod-5(n822)* into wild strains using CRISPR-*Cas9* gene editing (*Figure 8*)

To alter endogenous serotonin levels (*Ranganathan et al., 2001*; *Dempsey et al., 2005*; *Kullyev et al., 2010*), we introduced a point mutation corresponding to the *mod-5(n822)* loss-of-function allele using CRISPR-*Cas9* genome editing. This mutation changes cysteine 225 (codon TGT) to an opal stop codon (TGA) (*Ranganathan et al., 2001*). Gene editing was performed according to previously described procedures (*Ben Soussia et al., 2019*). In brief, the in vitro-synthetized crRNA crNB040 (GAAGUUCCGUGGGCGUCAUG) was used to target purified *Cas9* protein to the *mod-5* locus. The single-strand DNA oligonucleotide oNB322 (TCTAGATATTGTCACGTTGAGGTCATCTGAGC ATCTCGGaGTgTTCCAgGGgTTtgCtCATGACGCCCACGGAACTTCGGAATCCCAAATTTTctgaaat tttattgataaaattgaacaa) was co-injected as a repair template with the *Cas9* complex to introduce the Cys225Opal nonsense mutation. oNB322 also carries silent polymorphisms used for PCR detection using the oNB325/oNB328 (GCGTCATGaGcaAAcCCc/CAGACGACTGTGGACCCTTC) nucleotide pair. Final sequence validation was performed by Sanger sequencing on homozygous lines using the oNB327/oNB328 (ATCATCGCTCAAGCCGTCTA /CAGACGACTGTGGACCCTTC) primer pair.

## Determination of reproductive schedules and lifetime offspring production (*Figure 9*)

Lifetime production of (viable) offspring was analysed by isolating mid-L4 hermaphrodites onto individual NGM plates and transferring them daily to fresh NGM plates until egg-laying ceased. The number of live larvae was counted 24–36 hr after each transfer. Larvae in dead mothers were counted once they had exited the maternal body. Alternatively, mothers were squashed between slide and coverslip to count offspring in the uterus. All strains were scored in parallel.

## Analysis of lifespan and internal hatching (*Figure 10*)

Young adult hermaphrodites (mid-L4 +24 hr) were transferred to NGM plates seeded with fresh *E. coli* OP50. Individuals were transferred daily onto fresh plates until the end of egg laying, and every 3 days afterwards. Animals were scored as dead if they failed to respond to gentle touch with a platinum wire (N=90–102 individuals per strain, except for QG2873 with N=60).

## Quantification of hermaphrodite self-sperm number (*Figure 11—figure supplement 1*)

The total number of self-sperm was quantified in age-synchronized young adult hermaphrodites of the JU2593 strain containing between one to six eggs in utero. A single, usually the anterior, spermatheca was imaged at ×60 magnification by performing Z-sections (1 μm) covering the entire gonad (*Gimond et al., 2019*). Sperm number was determined by identifying condensed sperm nuclei visible in each focal plane using the Fiji plugin Cell Counter (*Schindelin et al., 2012*). For a given individual, sperm count was assessed for a single gonad arm and then multiplied by two to extrapolate the total sperm count. (N=19 individuals). In parallel, a cohort of individuals in the same experiment was used to count lifetime offspring production.

## Effects of internal hatching on physical integrity of the germline and scoring the presence of sperm (*Figure 11*, *Figure 11—figure supplement 1*)

Gravid hermaphrodites were fixed in methanol at different stages of reproductive adulthood and stained with DAPI to check for germline damage and the potential concomitant presence of sperm, unfertilized oocytes, and hatched larvae. Animals were fixed in cold methanol (–20 °C) for at least 30 min, washed three times with PBTw (PBS: phosphate-buffered saline, 137 mM NaCl, 2.7 mM KCl, 10 mM Na2HPO4, 2 mM KH2PO4, pH 7.4 containing 0.1% Tween 20) and squashed on a glass slide with Vectashield mounting medium containing 4,6-diamidino-2-phenylindole (DAPI; Vector

Laboratories, Newark, California, USA). Observations were performed using an Olympus BX61 microscope (N=45–193 individuals per strain).

## Measuring the time until hatching of laid eggs (*Figure 12B*)

For each strain, 10–20 adult hermaphrodites (mid-L4 +30 hr) were allowed to lay eggs for one hour on NGM plates seeded with *E. coli* OP50 and then removed from plates. The initial number of laid eggs was counted (N=30–170 eggs per strain) and the proportion of non-hatched eggs was determined every 60 min until all eggs had hatched. Mean values represented as bars correspond to the time at which 50% of eggs have hatched.

## Measuring egg-to-adult developmental time of laid eggs (*Figure 12C*)

For each strain, 10–20 adult hermaphrodites (mid-L4 +30 hr) were allowed to lay eggs for one hour on NGM plates seeded with *E. coli* OP50 (N=77–318 eggs per strain) and then removed from plates. Starting at 40 hr after egg laying, developmental progression of resulting progeny was surveyed using a high-resolution dissecting microscope to determine age at reproductive maturity, defined as the time point at which an animal had one to two eggs in the uterus. Animals that had reached reproductive maturity were progressively eliminated from the plate. Populations were scanned for mature adults every 1–2 hr until all individuals had reached reproductive maturity, that is for a maximum period of 16 hr. Mean values shown in *Figure 12C* correspond to the time at which 50% of individuals have reached maturity.

## Short-term competition assays (*Figure 12D and E*)

Short-term competition experiments of JU1200$_{WT}$ and JU1200$_{KCNL-1\ V530L}$ against a GFP-tester strain with genotype starting frequencies of 50:50 were carried out in parallel (*Figure 12D*). The two strains were competed separately against the GFP-tester strain PD4790, containing an integrated transgene [mls12 (*myo-2*::GFP, *pes-10*::GFP, *F22B7.9*::GFP)] in the N2 background, expressing green fluorescent protein (GFP) in the pharynx (*Figure 12D*). For each replicate, 20 laid eggs of either genotype were mixed with 20 laid eggs of the GFP-tester strain on a NGM plate; eggs were derived from a one-hour egg-laying window of adult hermaphrodites (midL4 +30 hr). After 4–5 days (when food became exhausted), the fraction of GFP-positive adult individuals was estimated by quantifying the fraction of GFP-positive individuals among a subpopulation of ~200–400 individuals per replicate using a fluorescence dissecting microscope. N=10 replicates per genotype.

Direct competition between JU751 (containing the V530L KCNL-1 variant) and other wild strains (JU651, JU660, JU814, JU820, JU822) isolated from the same compost substrate in Le Perreux-sur-Marne, France (*Barrière and Félix, 2007*; *Billard et al., 2020*; *Vigne et al., 2021*; *Fausett et al., 2022*; *Figure 12E*). For each the five strains, 20 freshly laid eggs were mixed with 20 freshly laid eggs from JU751 on a NGM plate; eggs were derived from a 1-hr egg-laying window of adult hermaphrodites (midL4 +30 hr). After 4–5 days (when food became exhausted), all adults present on each plate were transferred to fresh plates to allow for growth for 24 hr, after which the strong egg retention phenotype of JU751 was used to infer genotype frequencies (*Vigne et al., 2021*). N=10 replicates per strain.

## Effect of strong egg retention on progeny protection in response to strong environmental stress (*Figures 13 and 14*)

We examined the differences in survival of eggs and/or larvae developing ex utero versus in utero in response to different environmental stressors. For ex utero exposure, 10 synchronized mid-L4 +36 hr gravid mothers were dissected to extract eggs, which were transferred together to a single spot next to (but not directly on) the bacterial lawn on a single fresh NGM plate. A drop (20 microlitres) of M9 (control) or M9 containing ethanol or acetic acid in indicated concentrations was then deposited directly on eggs, which was absorbed by the agar in ~15 min. For in utero exposure (in parallel to ex utero exposure), 10 synchronized mid-L4 +36 hr gravid mothers were deposited into a drop (20 microlitres) of M9 containing ethanol or acetic acid in indicated concentrations (which was absorbed by the agar in ~15 min). Mothers were killed within a few minutes; no M9 control treatment as live adults can leave the drop rapidly. Forty-eight hr after the treatment, the number of live larval offspring was

counted. For each strain per treatment (ex utero versus in utero), 5–10 replicates were established (*Figure 13A and B*, *Figure 13—figure supplement 1*, *Figure 14A*, *Figure 14—figure supplement 1*). For experiment shown in *Figure 14B and C*, single mothers (instead of 10) were picked to individual plates (N=10 per treatment). For the experiment shown in *Figure 14C*, we used older hermaphrodites (mid-L4 +48 hr) containing larger numbers of internally hatched larvae to compare larval survival ex utero (extracted by dissection) versus in utero (larvae retained in mothers).

## Effect of strong egg retention on progeny development and reproduction in response to mild environmental stressors (*Figure 15*)

Eggs from adult JU751 hermaphrodites (mid-L4 +36 hr) or L1 larvae from mid-L4 +48 hr mothers were exposed either directly (ex utero) or indirectly (in utero) to mild environmental stressors: hyperosmotic stress (0.3M to 1M NaCl) (*Figure 15A and B*) and 1 M acetic acid (*Figure 15C and D*). Eggs were then allowed to hatch and develop for 45 hr, at which time we determined their developmental stages. To calculate differences in developmental maturation, we used a semi-quantitative scoring system (*Poullet et al., 2016*) to assign each developmental stage a specific index as follows: L3=1; early mid-L4=2; midL4=3; lateL4=4; Adult = 5. Lifetime offspring production of individuals descending from eggs and L1 larvae that had survived environmental insult and reached maturity was assayed as described in the material and methods section for *Figure 9*.

## Genome-wide association mapping

GWA mapping was performed using the NemaScan pipeline (*Cook et al., 2017*; *Widmayer et al., 2022*; *Crombie et al., 2023*).

## Local haplotype analysis

A neighbour-joining network of a region extending 1 Mb on either side of *kcnl-1* gene were established based on the VCF file dataset of CeNDR release 20220216 including 15 focal strains of a total of 48 isotypes. The region flanking the *kcnl-1* gene was extracted using the vcftools command line and the 48 isotypes from were then extracted using vcfselectsamples command line from galaxy (https://usegalaxy.org). The distance matrix and unrooted neighbour-joining trees were generated using the vk phylo tree nj command-line from the VCF-kit tools (*Cook et al., 2017*; *Widmayer et al., 2022*). Tree figures were made using the software iTOL (v5.6.3) (*Letunic and Bork, 2019*).

## Statistical analyses

Statistical tests were performed using R (*R Development Core Team, 2018*) and JMP 16.0 software. Data for parametric tests were transformed where necessary to meet the requirements for ANOVA procedures. Box-plots: The median is shown by the horizontal line in the middle of the box, which indicates the 25th to 75th quantiles of the data. The 1.5 interquartile range is indicated by the vertical line.

## Acknowledgements

*C. elegans* strains and substrate samples were kindly provided by the *Caenorhabditis* Natural Diversity Resource (*Cae*NDR), Marie-Anne Félix and Jean-Antoine Lepesant. We thank all students and technicians, including Céline Ferrari, Alex Lassagne, Ghada Bouzouida, Nicolas Schwartz, Matthieu Duval, for contributing to egg retention measurements. We thank the team of Erik Andersen, particularly Sam Widmayer, for helping with GWA mapping procedures. For helpful discussion and comments, we thank Kevin M Collins, Henrique Teotónio and the reviewers and editors. We would also like to thank *Cae*NDR (https://caendr.org) and WormBase (https://wormbase.org/) for providing resources and tools without which the analyses performed here would not have been possible.

# Additional information

## Funding

| Funder | Grant reference number | Author |
|---|---|---|
| Agence Nationale de la Recherche | ANR-22-CE13-0030-02 | Thomas Boulin<br>Christian Braendle |
| Agence Nationale de la Recherche | ANR-17-CE02-0017 | Christian Braendle |
| Agence Nationale de la Recherche | ANR-15-IDEX-01 | Laure Mignerot<br>Christian Braendle |
| Centre National de la Recherche Scientifique | | Laure Mignerot<br>Clotilde Gimond<br>Lucie Bolelli<br>Charlotte Bouleau<br>Asma Sandjak<br>Thomas Boulin<br>Christian Braendle |
| Université Côte d'Azur | | Laure Mignerot<br>Clotilde Gimond<br>Lucie Bolelli<br>Charlotte Bouleau<br>Asma Sandjak<br>Christian Braendle |
| Institut National de la Santé et de la Recherche Médicale | | Thomas Boulin<br>Christian Braendle |

The funders had no role in study design, data collection and interpretation, or the decision to submit the work for publication.

## Author contributions

Laure Mignerot, Conceptualization, Data curation, Formal analysis, Investigation, Visualization, Methodology, Writing – original draft, Writing – review and editing; Clotilde Gimond, Conceptualization, Formal analysis, Validation, Investigation, Visualization, Methodology, Writing – original draft, Writing – review and editing; Lucie Bolelli, Charlotte Bouleau, Asma Sandjak, Investigation, Methodology; Thomas Boulin, Investigation, Methodology, Writing – review and editing; Christian Braendle, Conceptualization, Resources, Formal analysis, Supervision, Funding acquisition, Validation, Investigation, Methodology, Writing – original draft, Project administration

## Author ORCIDs

Laure Mignerot (ID) http://orcid.org/0000-0003-3306-6244
Thomas Boulin (ID) http://orcid.org/0000-0002-1734-1915
Christian Braendle (ID) http://orcid.org/0000-0003-0203-4581

Reviewer #1 (Public Review): https://doi.org/10.7554/eLife.88253.3.sa1
Reviewer #2 (Public Review): https://doi.org/10.7554/eLife.88253.3.sa2
Author Response https://doi.org/10.7554/eLife.88253.3.sa3

# Additional files

## Supplementary files
• Supplementary file 1. A table of strains used in this study.
• MDAR checklist

## Data availability

All data generated or analysed in this study are included in the manuscript and supporting files. Source data files have been provided for all figures.

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
