## [Editor Report · eLife assessment]

This **important** work provides a thorough and detailed analysis of natural variation in *C. elegans* egg-laying behavior. The authors present **convincing** evidence to support their hypothesis that variations in egg-laying behavior are influenced by trade-offs between maternal and offspring fitness. This study establishes a framework for elucidating the molecular mechanisms underlying this paradigm of behavioral evolution.

---

## [Referee Report · Reviewer #1 (Public Review)]

Mignerot et al. performed a Herculean effort to measure and describe natural variation in *C. elegans* egg-laying behavior and egg retention. The paper is well written and organized. The authors show wild strains vary in egg retention with some extremes that appear phenotypically similar to species with viviparity (or live birth / internal hatching of offspring). They previously published a rare variant in the gene kcnl-1 that plays a role in egg retention but identify common variants in this study. They classify wild strains based on egg-retention to separate out the extremely different isolates. Egg laying has been extensively studied in the laboratory strain N2, but rarely addressed in natural strains. The authors investigate egg-laying behaviors using standard assays and find that their classified egg-laying groups have differences in sub-behaviors suggesting diverse roles in the ultimate egg-laying output. Then, they turn to the egg-laying circuit using both exogenous serotonin (5-HT), 5-HT modulatory drugs (e.g. SSRIs), and even genome editing to test epistasis with the mod-5 5-HT reuptake. The effects of 5-HT modulation and mutants are not predictive based on the basal behaviors and egg-retention phenotypes with the most extreme egg-retention strains differing in their responses. Interestingly, strains with more egg retention have decreased fitness (in their laboratory) measures but also provide a protective environment for offspring when exposed to common "natural" stressors. Their final conclusion that egg retention could be a trade-off between antagonistic effects of maternal vs. offspring fitness is supported well and sets the stage for future mechanistic studies across Caenorhabditis.

---

## [Referee Report · Reviewer #2 (Public Review)]

Mignerot et al. study variations in egg retention in a large set of wild *C. elegans* strains use detailed analysis of a subset of these strains to those that these variations in egg retention appear to arise from variations in egg-laying behavior. The authors then take advantage of the advanced genetic technology available in *C. elegans*, and the fact that the cellular and molecular mechanisms that drive egg-laying behavior in the N2 laboratory strain of *C. elegans* have been studied intensely for decades. Thus, they demonstrate that variations multiple genetic loci appear to drive variations in egg laying across species, although they are unable to identify the specific genes that vary other than a potassium channel already identified in a previous study from some of these same authors (Vigne et al., 2021). Mignerot et al. also present evidence that variations in response of the egg-laying system to the neuromodulator serotonin appear to underlie variations in egg-laying behavior across species. Finally, the authors present a series of studies examining how the retention of eggs in utero affects the fertility and survival of mothers versus the survival of their progeny in a variety of adverse conditions, including limiting food, and the presence of acute environmental insults such as alcohol or acid. The results suggest that variations in egg-laying behavior evolved as a response to adverse environmental conditions that impose a trade-off between survival of the mothers versus their progeny.

Strengths:

The analysis of variations in egg laying by a large set of wild species significantly extends the previous work of Vigne et al. (2021), who focused on just one wild variant strain. Mignerot find that variations in egg laying are widespread across *C. elegans* strains and result from changes in multiple genetic loci.

To determine why various strains vary in their egg-laying behavior, the authors take advantage the genetic tractability of *C. elegans* and the huge body of previous studies on the cellular and molecular basis of egg-laying behavior in the laboratory N2 strain. Since serotonin is one signal that induces egg laying, the authors subject various strains to serotonin and to drugs thought to alter serotonin signaling, and they also use CRISPR induced gene editing to mutate a serotonin reuptake transporter in some strains. The results are largely consistent with the idea that variations across strains alter how the egg-laying system responds to serotonin.

The final figures in the paper presents a far more detailed analysis than did Vigne et al. (2021) of how variations in egg retention across species can affect fitness under various environmental stresses. Thus, Mignerot et al. look at competition under conditions of limiting food, and response to acute environmental insults, and compare the ability of adults, in utero eggs, and ex vivo eggs to survive. The results lead to an interesting discussion of how variations in behavior result in a trade-off in survival of mothers versus their progeny. The authors in their Discussion do a good job describing the challenges in interpreting the relevance of these laboratory results to the poorly-understood environmental conditions that *C. elegans* may experience in the wild. The Discussion also had an excellent section about how the ability of a single species to strongly regulate egg-laying behavior in response to its environment, and how this ability could be adaptive. Overall, these were the strongest and most interesting aspects of Mignerot et al.

Weaknesses

The specific potassium channel variation studied by Vigne et al. (2021) has by far the strongest effect on egg laying seen in the Mignerot et al. study and remains the only genetic variation that has been molecularly identified. So, Mignerot et al. were not able to identify any additional specific genes that vary across species to cause changes in egg laying, and this limited their ability to generate new insights into the specific cellular and molecular mechanisms that have changed across species to result in changes in egg laying behavior.

The authors' use of drug treatments and CRISPR to alter serotonin signaling yielded some insights into mechanistic variations in how the egg-laying system functions across strains, but these experiments only allow very indirect inferences into what is going on. The analysis in Figures 4 and 5 generates a complex set of results that are not easy to interpret. The clearest result seems to be that strains carrying the KCNL-1 point mutation lay eggs poorly and exogenous serotonin inhibits rather than stimulates egg laying in these strains. This basic result was to a large extent reported previously in Vigne et al. 2021.

The analysis of how differences between strains mechanistically result in changes in egg-laying behavior and egg retention, while excellent in concept, is only modestly successful. The analysis of the temporal pattern egg-laying behavior in Figure 3B-3F is relatively weak. Whereas the state of the art in analyzing this behavior is to take videos of animals and track exactly when they lay eggs, analyzing 40 or more hours of behavior per strain, the authors used a lower-tech method of just examining how many eggs were laid within 5-minute intervals over a period of just three hours per strain. While this analysis was sufficient to demonstrate some statistically significant differences in the pattern of egg laying in some strains, it is unclear to what extent these differences could be sufficient to explain the differences in accumulation of unlaid eggs between these strains. In contrast, the variations in age of the onset of egg-laying behavior in Fig 3G and 3H between strains were very strong and may be more likely to reflect mechanistic differences in how egg laying is controlled that could result in the differences in retention of unlaid eggs seen among the strains tested. In the Discussion, the authors extensively write about the work of the Collins lab showing that retained eggs stretch the uterus to produce a signal that activates egg-laying muscles. Could it be that really this mechanism is the main one that varies between strains, leading to the observed variations in time to laying the first egg as well as variations in the number of retained eggs throughout adulthood?

---

## [Author Response]

The following is the authors’ response to the original reviews.

**Reviewer #1:**
1. Can the authors statistically define the egg-laying classes? In some parts of the manuscript, the division between the different classes could be more ambiguous. I understand that the class III strains are divided by the kcnl-1 genotype, but given the different results for diverse traits, it could be more clear to keep them as one class. Also, overall, the authors choose a collection of 15 strains across the different classes to phenotype for many traits and perform genome edits. It is understandable that they cannot test all strains, but given the variation across traits and classes, it might be good to add a few more caveats about how these strains might not be representative of all strains across the species.

Response: The egg-laying classes were defined as in Figure 1A by arbitrarily chosen cut-offs (at 10, 10-25, and 25 eggs in utero) to simplify subsequent analyses. We added this explanation to the first paragraph of the results section. However, the differences in average egg retention are significantly different between the four defined classes using the 15 selected strains (Fig. 2A).

We think that the distinction between Class IIIA and IIIB strains is important and justified because the two Classes significantly differ in mean egg retention (Fig. 2A) and because Class IIIB harbour the large-effect variant KCNL-1 V530L whereas Class IIIA do not.

We agree that the 15 selected strains are not necessarily representative of all strains across the species. We have added a note of caution regarding this point to the first paragraph of the section “Temporal progression of egg retention and internal hatching”: “Note that this strain selection, especially concerning the largest Class II, is unlikely to reflect the overall strain diversity observed across the species". In addition, we have reworded the first sentence of this paragraph as follows: “ To better characterize natural variation in *C. elegans* egg retention, we focused on a subset of 15 strains from divergent phenotypic Classes I-III, with an emphasis on Class III strains exhibiting strong egg retention (at mid-L4 + 30h) (Fig. 2A and 2B).”

2. For the GWAS experiments, the authors should describe if any of the QTL overlap with hyper-divergent regions in the strain set. The QTL could be driven by these less well defined regions.

Response: We have added the following sentence: “The three QTLs do not align with any of the recently identified hyper-divergent regions of the genome (Lee et al., 2021).

3. The authors should look at correlations between the mod-5(n822) edit phenotypes and the exogenous 5-HT and SSRI phenotypes to demonstrate how the traits can differ. Some correlation plots might help that point as well.

Response: We examined all possible correlations as suggested: none are significant and strain effects on trait differences are idiosyncratic, as written in our results section. The correlational analyses remain of limited value due to small samples: N=10 for mean strain values for measured phenotypes. We therefore feel that these analyses do not provide any additional insights beyond our figures (4C, 4D, 5C, 5D, S5A-C ) and our statement on page 15: “As in previous experiments (Fig. 4C and 5C), we find again that strains sharing the same egg retention phenotype may differ strongly in egg-laying behaviour in response to modulation of both exo- and endogenous serotonin levels (Class IIIA: ED3005 and JU2829) (Fig. 5D and S5C).”

4. Figure 6D, was there any censoring of the data? Normally, these types of studies are plagued by an increase in censored animals that can decrease significance. The effects among the classes seem large, but statistical comparisons might help as well.

Response: There was no censoring of animals (censoring of animals in lifespan studies is usually done by removing “bags of worms”, which here was our study phenotype). We now mention this in the corresponding figure legend. We also added a statistical analysis showing that mean survival was significantly different between all Classes.

5. Many of the traits, edits, and deeper analyses are performed on the JU751 genetic background. This choice is sensible, otherwise, the work can increase exponentially. However, the authors should add a caveat about how these results might be limited to JU751 and other strains might respond differently.

Response: For certain experiments, it was not feasible to include multiple strains from all phenotypic classes, so we selected JU751 (Class IIIB) and JU1200 (Class II), for which we had established CRISPR-engineered lines to modulate the egg retention phenotype by a single amino acid change in KCNL-1. To emphasize that these experimental observations cannot be generalized, we added the following statement in the relevant results section: “These experimental results offer preliminary evidence (bearing in mind that our analysis was primarily centered on a single genetic background) that laying of advanced-stage embryos may enhance intraspecific competitive ability, particularly in scenarios where multiple genotypes compete for colonization and exploitation of limited, patchily distributed resources.”

6. The authors argue that evolution could be acting on specific parts of the egg-laying machinery (e.g., muscledirected signaling components). It might be useful to look at levels of standing variation and selection at groups of loci compared to genomic controls to see if this conclusion can be strengthened.

Response: This is a good idea but how to select pertinent candidate loci is unclear (there are over 300 genes with effects on egg laying, http://www.wormbase.org). In addition, the genetics of muscle-directed signalling components in egg laying is only starting to be explored, with no specific candidate genes having been identified (Medrano & Collins, 2023, Curr Biol). We therefore think that such an analysis is currently not possible.

7. Completely optional: The authors present a compelling and interesting case for transitions and trade-offs between oviparity and viviparity. The C. vivipara species has a different egg-laying mode than other Caenorhabditis species. The authors could add a short section describing their expectations about the neuronal morphology, 5-HT circuits, and muscle function in this species given their results. What genes or circuits should be the focus of future studies to address this question in Caenorhabditis. Also, Loer and Rivard present some similar ideas based on the differences in 5-HT staining neurons across diverse nematodes. Those results can be incorporated and discussed as well.

Response: Our current research focuses on the evolution of egg laying in different Caenorhabditis species. So far, however, it remains difficult to provide specific hypotheses on how the egg-laying circuit has changed in C. vivipara. We rephrased the final paragraph of the discussion to incorporate some of the reviewer’s suggestions: “Nematodes display frequent transitions from oviparity to obligate viviparity in many distinct genera (Sudhaus, 1976; Ostrovsky et al., 2015), including in the genus Caenorhabditis, with at least one viviparous species, C. vivipara (Stevens et al., 2019). Although evidence exists for the evolution of egg-laying circuitry across oviparous Caenorhabditis species (Loer and Rivard, 2007), the specific cellular and genetic changes responsible for the transition to obligate viviparity in C. vivipara have yet to be examined. Resolving the genetic basis of intraspecific variation in *C. elegans* egg retention, including partial or facultative viviparity, may thus shed light on the molecular changes underlying the initial steps of evolutionary transitions from oviparity to obligate viviparity in invertebrates.”

Specific edits:1. Perhaps a silly point, but "parity" (to my knowledge) does not have a biological meaning on its own. I suggest "egg-laying mode" or "birth mode".

Response: This term has been used previously in the literature(e.g.https://onlinelibrary.wiley.com/doi/10.1111/jeb.13886 or https://doi.org/10.1101/2023.10.22.563505). However, as the referee rightly points out, this is not a standard term. We therefore replaced “parity mode” with “egg-laying mode”.

2. "Against fluctuating environmental fluctuations" is a bit strange

Response: Corrected.

3. The first publications of Egl mutants were by the Horvitz lab so some citations are not in all of the first descriptions of the trait (early in Results)

Response: We have added the relevant work (Trent 1982, Trent 1983, Desai & Horvitz 1989) to this paragraph in the early results section.

4. "Strong egg retention usually strongly..." is a bit strange

Response: Corrected.

5. Figure 8G font looks smaller than the others.

Response: Corrected.

**Reviewer #2:**
1. In Figure 1A, I infer that in the graph class I measurements are represented by dark blue dots and class II by purple dots. I am having a really hard time distinguishing between these two colors in the graph. In the pie chart I have no problem, but in the graph the black lines around the colored dots seem to obscure the colors. Not sure how to fix this graphical problem, but it is preventing the graph from communicating the results effectively.

Response: We have changed the colours, spacing and format of this figure to resolve this problem.

2. The behavioral analysis of Figure 3B-3F is problematic. The experimental methods used and the interpretation of the results each have issues. This is cause for concern since this is the most direct analysis of the actual variations in egg-laying behavior across strains presented in this paper.This experiment is modeled after the work of Waggoner et al. 1998, who recorded egg laying events of individual worms on video over several hours and noted the exact time of individual egg laying events. Waggoner et al. found in the reference *C. elegans* strain N2 that egg-laying events occurred in ~2 minute clusters ("active phases") separated by ~20 minute silent periods ("inactive phases"). Mignerot et al. did not take continuous videos of animals, but rather examined plates bearing a single worm only every 5 minutes and noted the number of new eggs that appeared on the plate in each 5-minute interval. From these data, the authors claim they have measured the intervals between "egg-laying phases" (the term used in the Figure 3 legend). In the Results, the authors explicitly claim they are measuring the timing and frequency of actual active and inactive egg-laying phases. Apparently, all the eggs laid within one 5-minute interval are considered to have been laid in a single active phase, and the time between 5-minute intervals containing egg laying events is considered an "inactive phase" and is measured only with a resolution of 5 minutes. It is not explained anywhere how the authors handle the situation of seeing eggs laid in two consecutive 5-minute intervals. Is that one active phase that is 10 minutes long, or is that two separate active phases with a 5-minute active phase in between? Because of this ambiguity in how they define active and inactive phases, I find it impossible to understand and judge the data presented in Fig. 3D-3F. The authors in the results state that "Class I and Class IIIB displayed significantly accelerated and reduced egg laying activity respectively (Fig. 3C to 3E)" . I assume they are referring to the statistical analysis described in the figure legend, which is quite difficult to understand. Frankly, just looking at the graphs in Fig. 3D3F, it is hard for the reader to identify specific features shown in the graphs can explain why, for example, Class I strains have fewer retained eggs than Class III strains. So, I found this analysis very unsatisfying.I also feel the authors are making an unwarranted assumption that their non-N2 strains will have distinguishable active and inactive phases of egg-laying behavior analogous to those seen in the N2 strain. Given the possibly large variations in egg-laying behavior in the various strains examined, that assumption should be questioned. Thus, framing the entire analysis of behavior patterns in terms of the length of active and inactive phases might not be appropriate.

Response: This comment validly highlights important problems and limitations of our scan-sampling method to quantify strain differences in egg-laying behaviour. We acknowledge that we failed to present the data with due diligence, and clarity regarding terminology and interpretation. However, we think that some of these results are still of value after revised presentation. Our biggest mistake was to use the terms “active and inactive phase”, as coined by Waggoner et al. 1998. We are aware that our measures are not equivalent to these previously defined measures but have been sloppy with terminology. We therefore carefully reworded this entire results section, using clear definitions to indicate differences between the Waggoner assay and our assay (including a graphical representation of our assay design in the revised Fig. 3B). In brief, our simplified assay is useful to estimate the frequency and approximate duration of prolonged inactive periods of egg laying because we can unambiguously determine intervals in which eggs were laid or not. In contrast, as pointed out by the reviewer, we cannot determine if multiple active phases occurred within a 5-min interval, nor can we estimate the duration of an active “phase”. We now state this limitation explicitly in the manuscript. What our results do show is that the number of intervals during which egg laying occurred is significantly different between strains and Classes: Class I (low retention) have a higher number of intervals with egg-laying events, whereas Class IIIB showed a reduced number of such events (Fig. 3D). We can therefore also roughly estimate the mean time (per individual) between two egg-laying intervals, giving us a proxy for prolonged periods when egg-laying is inactive (Fig. 3E); we note that our estimate for N2 is very close to what has been previously measured (~20 min). Therefore, we can confidently conclude that there are natural strains which have both shorter (Class I) and longer (Class IIIB) inactive periods of egg laying. These results partly align with observed variation in egg retention. However, we agree with the reviewer – as we had stated both in results and discussion sections – that these behavioural differences act together with differences in the sensing of egg accumulation in utero (as suggested by results shown in Fig. 3G and 3H). We also agree that it seems very plausible that the observed behavioural differences, as revealed by scan-sampling, may only have a secondary role in accounting for natural variation in egg retention. We will be testing these hypotheses specifically in our future research.

Note: The statistical analyses are nested ANOVAs to ask (a) does the value differ between strains within a given class and (b) does the value differ between Classes? Classes labelled with different letters in the figures therefore significantly differ in their mean values, demonstrating that measured behavioural phenotypes consistently differ between some (but not all) phenotypic classes, yet largely in line with their egg retention phenotypes (Fig. 3D and 3E).

3. Figure 4A is a schematic diagram of how the egg-laying circuit works based on previous literature, and the authors cite Collins et al. 2015 and Kopchock et al. 2021 as their sources. One feature of this figure seems unwarranted, namely the part indicating that egg accumulation acts on the UM muscles, and the statement in the legend that "mechanical excitation of uterine muscles (UM) in response to egg accumulation favours exit from the inactive state (Collins et al., 2016)". I believe Collins et al. 2016 showed that egg accumulation favors egg laying and may have speculated that it does so by stretching the um muscles, but this idea remains speculative and has not been established by any experimental data. I point out this issue,in particular, because it may bear on the nice data the authors of this manuscript show in Figure 3G and 3H, which show that some strains accumulate many eggs in the uterus before they initiate egg laying.

Also, in Figure 4A and 4B, the legend does not explain the logic of the green areas labeled "egg-laying active phase" and the yellow area labeled "egg-laying inactive state". I was not sure what sure how to interpret these features of the graphics.

Response: The input from uterine muscles remains indeed hypothetical, and we have corrected the figure accordingly, now simply referring to the feedback of egg accumulation on egg laying activity, as recently characterized in more detail by Medrano & Collins (2023, Curr Biol).

The green/yellow backgrounds shown in figures 4A (and 4B) are not useful and we have removed them.

5. Results, page 11: "We used standard assays, in which animals are reared in liquid M9 buffer without bacterial food." In the standard assays, animals are reared on NGM agar plates with bacterial food, and then at the start of the egg-laying assay, are transferred to liquid M9 buffer without bacterial food. I assume that is what these authors did, and they should correct the language of the text to make it more accurate.

Response: The reviewer is correct. We have incorporated this change to improve accuracy.

6. The authors note that "serotonin induced a much stronger egg-laying responds in the Class IIIA strain ED3005 than in other strains (Fig. 4C)". I would like to point out to the authors that strains such as ED3005 that have a very large number of unlaid eggs in their uterus are prone to lay a very large number of eggs when treated with exogenous serotonin, simply for the trivial reason that they have more eggs to release. This was previously seen in, for example, in Desai and Horvitz (1989) in certain egg-laying defective mutants.

Response: This is an important point and our comparison of ED3005 to ALL other strains is problematic. We changed this result description by stating that ED3005 shows possible serotonin hypersensitivity compared to strains with similar levels of egg retention (Class IIIA): “In addition, serotonin induced a much stronger egg-laying response in the strain ED3005 than in other Class IIIA strains with similar levels of egg retention (Fig. 4B). ED3005 may thus exhibit serotonin hypersensitivity, which has been observed in certain egg-laying mutants where perturbed synaptic transmission impacts serotonin signalling (Schafer and Kenyon, 1995; Schafer et al., 1996).”

7. In Figure 4 the authors show that all strains lay eggs in response to fluoxetine and imipramine, but some strains (Class IIIB) do not lay eggs in response to serotonin. They then cite a series of papers, starting with Trent et al. 1983, that they claim show that this specific phenotype demonstrates that the HSN neurons are functionally releasing serotonin (bottom of page 11). This statement needs to be removed - it is incorrect. It is true that egg laying in response to fluoxetine and/or imipramine AS WELL AS egg laying in response to serotonin has been interpreted as indicating the presence of HSN neurons that functionally release serotonin to stimulate egg laying (these were referred to as Category C by Trent et al., 1983). However, the mutants that Mignerot et al. are talking about (those that don't respond to serotonin but do respond to imipramine/fluoxetine) were called Category D by Trent et al., 1983, and to my knowledge these have never been interpreted as necessarily having functionally intact HSN neurons. Mutants such as these that can lay eggs in some circumstances but cannot lay eggs in response to exogenous serotonin have usually been interpreted as having egg-laying muscles that are defective in responding to serotonin.How can we interpret strains that respond to imipramine/fluoxetine and not serotonin? Mignerot et al. cite some of the papers (Kullyev et al. 2010; Wenishenker et al., 1999; Yue et al., 2018) showing that imipramine and fluoxetene have off-target effects and can stimulate egg laying by acting through proteins other than the serotonin-reuptake inhibitor. The authors later in their discussion at the top of Page 24 also cite Dempsey et al 2005, a paper that also argues that imipramine and fluoxetene act via off target effects. However, currently in Figure 4B Mignerot et al. emphasize that the serotonin reuptake inhibitor is the target of these drugs. Since the results presented for Class IIIB strains are not in accord with this interpretation, this seems misleading to me. The bottom line for me is that class IIIB strains cannot respond to exogenous serotonin, but can lay eggs in other conditions, so perhaps there is something specifically wrong with their ability to respond to serotonin.

Response: We thank the reviewer for this important comment – we misinterpreted some of these past findings and our statements were either inexact or incorrect. We have revised this section accordingly: “Both drugs also stimulated egg laying in the Class IIIB strains and the Class IIIA strain JU2829 for which exogenous serotonin either inhibited egg laying or had no effect on it (Fig. 4B). In the past, mutants unresponsive to serotonin yet responsive to other drugs, including fluoxetine and imipramine, have been interpreted as being defective in the serotonin response of vulval muscles (Trent et al., 1983; Reiner et al., 1995; Weinshenker et al., 1995). This is indeed the likely case of Class IIIB strains carrying the KCNL-1 V530L variant thought to specifically reduce excitability of vulval muscles (Vigne et al., 2021). Our results therefore suggest that JU2829 (Class IIIA) may exhibit a similar defect in vulval muscle activation via serotonin caused by an alternative genetic change. Overall, these pharmacological assays do not allow us to conclude if and how HSN function has diverged among strains because the mode of action and targets of tested drugs has not been fully resolved. Nevertheless, our results are consistent with previous models proposing that these drugs do not simply block serotonin reuptake but can stimulate egg laying, to some extent, through mechanisms independent of serotonergic signaling (Trent et al., 1983; Desai and Horvitz, 1989; Reiner et al., 1995; Weinshenker et al., 1995, 1999; Dempsey et al., 2005;Kullyev et al., 2010; Branicky et al., 2014; Yue et al., 2018).”

We removed the oversimplified Fig. 4B to avoid any misinterpretation.

8. In Figure 7B and 7C, the authors should add some type of error bars to the graphs to and give the readers an idea of whether the differences between strains that they write about are statistically significant or not.

Response: These are frequency data to describe temporal dynamics of hatching (N=45-72 eggs per strain) (Fig. 7B) and development in single cohorts (N=48-177 eggs per strain) (Fig. 7C), hence, the absence of error bars.

We agree that this representation of the data is not very telling. We therefore changed the data representation in these two figures to show that there are clear, statistically significant, negative correlations between egg retention and time to hatching / egg-to-adult developmental time.

9. When the authors reference a list of papers in a single list, e.g. "(Burton et al., 2021; Fausett et al., 2021; Garsin et al., 2001; Padilla et al., 2002; Van Voorhies and Ward, 2000)" they seem to do so in alphabetical order by the first author's last name. I believe the usual practice is to list references by year of publication, with the earliest first.

Response: We corrected citation style according to eLIFE format.

10. At the top of page 24, the authors write "It seems unlikely, however, that any of these variants strongly alter central function of HSN and HSN-mediated signalling because fluoxetine and imipramine, known to act via HSN (Dempsey et al., 2005; Trent et al., 1983; Weinshenker et al., 1995), triggered a robust stimulatory effect on egg laying in all examined strains (Fig. 4C)." I believe that the Weinshenker paper in fact showed that imipramine does not act via the HSN, and the Dempsey paper suggested that both drugs can act at least in part independently of the HSN. Therefore, the authors should revise their statement.

Response: We have removed the sentence.

**Reviewing Editor:**
Minor suggestions:1. p. 2, fifth line from bottom: "lead" instead of "leads";1. p. 2, last line: "muscle" instead of "muscles";1. p. 3, first full paragraph, 17th line: "populations" instead of "population";1. p. 5, fourth line from bottom: Delete first comma;1. p. 6, Figure 1D: "of" instead of "off";1. p. 7, fifth line: "KCNL-1";1. p. 9, third paragraph, second line: please clarify "late mid-L4";1. p. 16, first line: "exogenous";1. p 20, first paragraph, beginning of second sentence: "Whether" instead of "If";1. p. 22, ninth line from bottom: delete "shaped by";1. p. 23, last paragraph, third and eighth lines from bottom: change "between" to "among"

Response: Thank you. All corrected.

Additional changes:

Figure 5A: We removed figure 5A showing a cartoon of mod-5/SERT and its effects on serotonin signalling. This figure was incorrectly showing that MOD-5 is expressed in HSN (Jafari et al 2011 J. Neuroscience, Hammarlund et al 2018 Neuron).

Abstract: We reworded the abstract to reduce its length.